# Research on a Real-Time Monitoring System for Campus Woodland Fires via Deep Learning

**Dengwei Xu** [1], **Jie Chen** [2], **Qi Wu** [2] **and Zheng Wang** [3,*]

1   Security Office, Nanjing Forestry University, Nanjing 210037, China; xudengwei@njfu.edu.cn
2   College of Mechatronic Engineering, Nanjing Forestry University, Nanjing 210037, China; 16670469734@163.com (J.C.); wuqi1236547895@163.com (Q.W.)
3   College of Materials Science and Engineering, Nanjing Forestry University, Nanjing 210037, China
*   Correspondence: wangzheng63258@163.com

**Abstract:** To solve the problems of low recognition accuracy and large amounts of computation required in forest fire detection algorithms, this paper, aiming to make improvements in these two aspects, proposes a G-YOLOv5n-CB forest fire detection algorithm based on the YOLOv5 algorithm and develops a set of real-time fire monitoring systems applicable to campus forest land with the aid of deep learning technology. The system employs an unmanned vehicle to navigate automatically and collect image information through a camera and deploys its algorithm on the unmanned vehicle's Jetson Nano hardware platform. The results demonstrate that the proposed YOLOv5n-CB algorithm increased the mAP value index by 1.4% compared with the original algorithm on the self-made forest fire dataset. The improved G-YOLOv5n-CB model was deployed on the Jetson Nano platform for testing, and its detection speed reached 15 FPS. It can accurately detect and display real-time forest fires on campus and has, thus, a high application value.

**Keywords:** campus forest fire detection; deep learning; YOLOv5; lightweight network; automatic navigation

## 1. Introduction

Forests in nature are a precious resource for humankind and a gift from the earth. However, forest fires not only cause damage to the natural environment but also seriously endanger human life and give rise to huge losses. Therefore, studying forest fire monitoring and early-warning systems is particularly important. Many Chinese universities have experimental woodlands on and off campus for their students to study and research. However, campus woodlands are usually located at remote sites; should a fire occur, it is less likely to be discovered in time and is prone to cause significant harm. Hence, developing a set of real-time monitoring systems for forest fire detection in campus woodlands is an urgent task.

The main content of a real-time fire monitoring system lies in applying a target detection algorithm and a forest fire detection algorithm. On the one hand, the research orientation of object detection has gradually shifted from traditional object detection algorithms to deep learning-based object detection algorithms in the past ten years. Many researchers have reaped good results in studying traditional object detection algorithms [1]. The directional gradient histogram algorithm proposed by Navneet Dalal and Bill Triggs in 2005 has proven to be a fundamental algorithm for traditional target detection [2]. And, with the rapid development of deep learning, the target detection performance of convolutional neural network structures has been greatly improved due to their remarkable abilities in feature extraction. In 2019, multi-scale target detection tasks with different perceptions were first undertaken in the TridentNet network structure proposed by Li et al., and the detection record of the COCO dataset was broken again [3]. In 2015, the YOLO algorithm proposed by Redmon et al. witnessed, for the first time, regression thought to be specifically

applied to the algorithm's target detection task [4], which adopted the idea of regression to realize the regression of candidate boxes while classifying objects instead of the concept of the candidate region. Based on Faster R-CNN, the algorithm improved the detection speed tenfold, making real-time detection possible. In 2020, Bochkovskiy et al. proposed YOLOv4, a master of the YOLO series of algorithms, to improve the YOLO algorithm in multiple stages [5]. On the other hand, forest fire detection methods may be divided into methods using traditional technologies and those based on deep learning algorithms. Traditional forest fire detection methods can be roughly divided into the following five categories: ground patrol, observation station detection, satellite remote sensing detection, sensor detection, and aerial patrol. As insurmountable difficulties in forest fire detection by traditional techniques exist, some researchers have begun to adopt classical deep learning models to implement forest fire detection tasks and have obtained satisfactory results. In 2019, Li et al. [6] applied the SqueezeNet network to a forest fire detection task, integrating multi-scale information while ensuring the integrity of the resolution information to attain a high detection effect. In 2020, Avula et al. [7] proposed a forest fire smoke detection method based on a fuzzy entropy optimization threshold in combination with a convolutional neural network. In 2021, Zhang Yong et al. [8] introduced MobileNetV3-Large into the YOLOv3 network, significantly reducing the model size. When the improved model was applied to the substation fire detection scene, the accuracy and detection speed were improved [9]. In 2022, PI Jun et al. [10] replaced the backbone network of the YOLOv5s network with the Shufflenetv2 network. They combined it with the idea of channel recombination to improve the efficiency of feature extraction and significantly accelerated the detection speed of the network. In 2022, Zhang Rong et al. [11] used GhostNet to extract features and combined them with the Fcos detection network to significantly reduce the number of parameters and the workload of computation, thus creating a lightweight fire detection network.

It is, thus, evident that applying deep learning to forest fire monitoring can improve the speed and accuracy of fire detection [12]. However, the accuracy of the current forest fire detection algorithm based on deep learning still needs to be improved, and its complex model and high computing demand make it difficult to meet the real-time requirements of forest fire detection. A typical forest fire detection system usually acquires images through drones or robots with onboard cameras [10]. Then, it sends the images to the high-performance computer deployed by the model for processing, making it difficult to synchronize video acquisition and image processing. To address this issue, we integrate deep learning and robotics in this work to achieve real-time forest fire detection. Through the field deployment of the detection algorithms on the main boards, robots can now be used not only for real-time image collection of a campus woodland but also to perform the lightweight fire detection model improved from the YOLOv5 algorithm for real-time fire detection in both a rapid and accurate fashion. This realization of real-time fire detection in the scenario of campus woodlands is significant for the research on and application of real-time forest fire detection based on deep learning.

## 2. Forest Fire Detection Algorithm Based on the Improved YOLOv5

### 2.1. Basic Structure of YOLOv5

YOLOv5 has five network structures, including YOLOv5n, YOLOv5s, YOLOv5m, YOLOv5l, and YOLOv5x. YOLOv5s is suitable for resource-constrained environments; YOLOv5m is the choice for balancing performance and speed; and YOLOv5l and YOLOv5x are ideal for tasks that require high precision. YOLOv5n is a compromise option that can perform better on some tasks. Due to the high real-time requirements for forest fire detection, an extensive network structure affects the detection speed. Therefore, the YOLOv5n network with a smaller network structure was selected as the benchmark network in this study. Its network structure is shown in Figure 1.

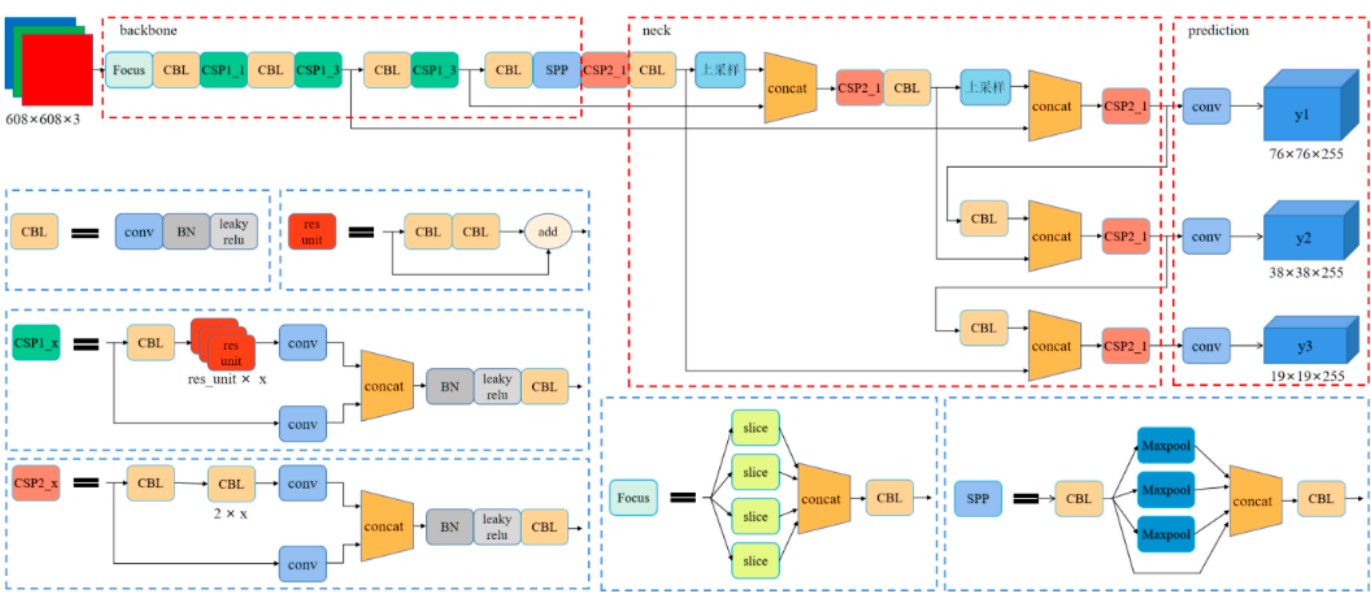

**Figure 1.** Network structure of YOLOv5.

In Figure 1, the YOLOv5 network structure mainly consists of four parts: input, backbone, neck, and output, and each part is composed of different essential components to accomplish its various functions.

### 2.2. The Improvement of the YOLOv5 Network

#### 2.2.1. Mix the Attention Mechanism

The idea of the neural network attention mechanism is derived from the human visual attention mechanism, which is to devote more attention to critical areas. At the same time, the background is filtered or ignored. The attention mechanism of the neural network is to imitate the human visual mechanism by using a set of weights to focus on the input's key information, obtain more useful feature information, and suppress or even filter irrelevant information. Incorporating a neural network attention mechanism can enable the model to extract more important features without increasing the calculation burden and the number of parameters and effectively improve the detection effect of the model. This paper employs SE, ECA, and CBAM attention mechanisms to improve the YOLOv5 network.

Among them, the SE attention mechanism helps solve the loss problem caused by the different importance of different channels in the feature graph during convolution pooling. Based on the original learning mechanism, it opens up a new network path, obtains the attention degree of each channel in the feature graph through operation, and assigns an attention weight to each feature channel according to the degree so that the convolutional network pays more attention to these feature channels, and then secures the channel of the feature graph that is useful for the current task and suppresses feature channels that are not useful. Its structure is shown in Figure 2.

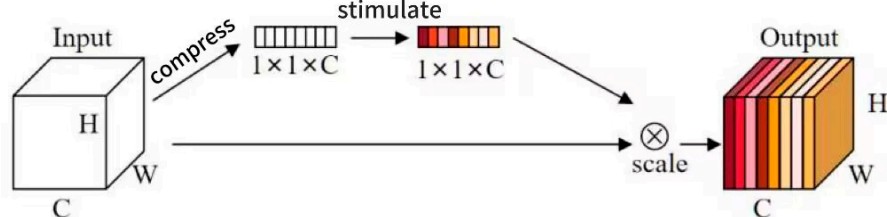

**Figure 2.** SE module structure diagram.

C: channel, which indicates the number of channels in an image.

H: height, which indicates the number of pixels in the vertical dimension of the image.

W: width, which indicates the number of pixels in the horizontal dimension of the image.

Based on the SE module, the ECA module changes the FC layer used in the SE module to $1 \times 1$ convolution to learn channel attention information. This operation avoids the reduction in channel dimension when learning channel attention information and effectively reduces the number of parameters. Its structure is shown in Figure 3.

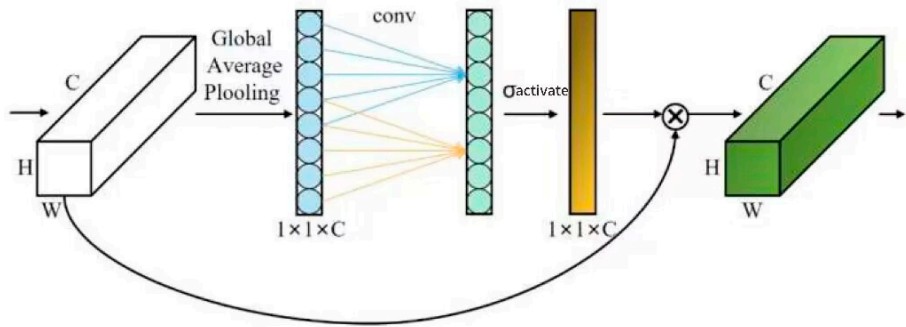

**Figure 3.** ECA module structure diagram.

The CBAM attention mechanism module is a combination of channel and spatial attention, which operates first through the former and then through the latter [13]. The weights obtained by the two attention modules multiplied by the corresponding location feature map element values are derived from the adaptive feature information. Compared with other attention mechanisms based on single attention, CBAM combines the two to achieve better results in model performance. Its structure is shown in Figure 4.

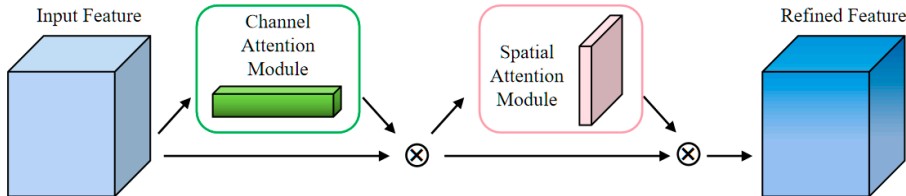

**Figure 4.** CBAM module structure diagram.

### 2.2.2. Integrate the BiFPN Structure

Based on the bidirectional path fusion of PANet, Google proposed the BiFPN feature fusion method. BiFPN boasts simultaneous access to bidirectional multi-scale connection and unique feature weighted fusion mode, which not only strengthens the fusion of different scale feature information between the deep layer and shallow layer but also adds horizontal correlation between the same scale feature information, thus effectively avoiding the loss of feature information caused by too deep a feature extraction network. Its structure is shown in Figure 5.

C indicates the input feature map before feature fusion, which can also be called the feature map of the current level. These feature maps will go through a series of feature fusion operations, including up-sampling, down-sampling, and feature fusion, and finally generate a fused feature map.

P represents the output feature map after feature fusion, also called the feature map after fusion. This feature map is then passed to the BiFPN structure at the next level for further feature fusion and processing.

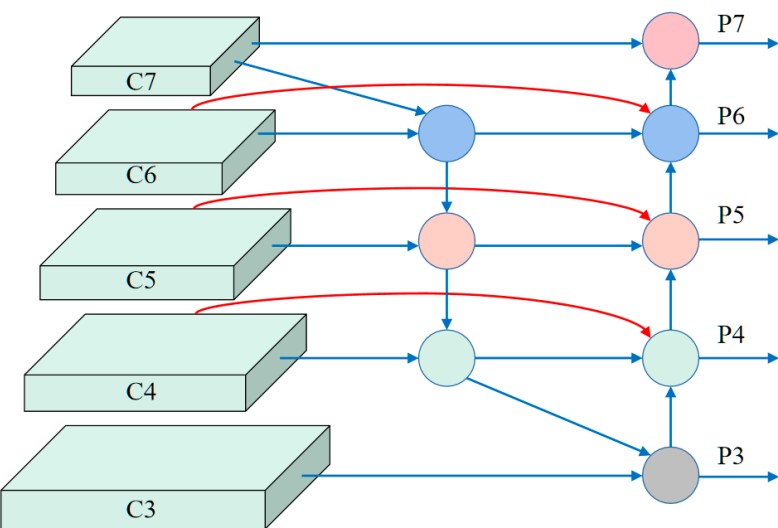

**Figure 5.** BiFPN structure diagram.

### 2.2.3. Lightweight Improvement Model Combined with GhostNet

Huawei Noah's Ark Laboratory proposed the lightweight network GhostNet in 2020, designing a unique convolutional structure that significantly reduces the amount of computation. The detection effect is slightly improved under the same amount of computation compared with other lightweight networks. The design idea comes from the visual analysis of the intermediate feature images generated in the process of deep learning, and it is found that the abundant and redundant information in the feature map can often reflect the comprehensiveness of the input data. Therefore, the researchers designed a low-cost and high-yield Ghost convolution unit to carry out these feature mappings containing redundant information to reveal the information behind the intrinsic features fully. The Ghost convolution unit adds the weight value of the feature map obtained by depth convolution to the input feature map to retain the context feature information better. GhostNet designs a unique Ghost module based on the Ghost convolution unit to extract features from input images. Its structure is shown in Figure 6.

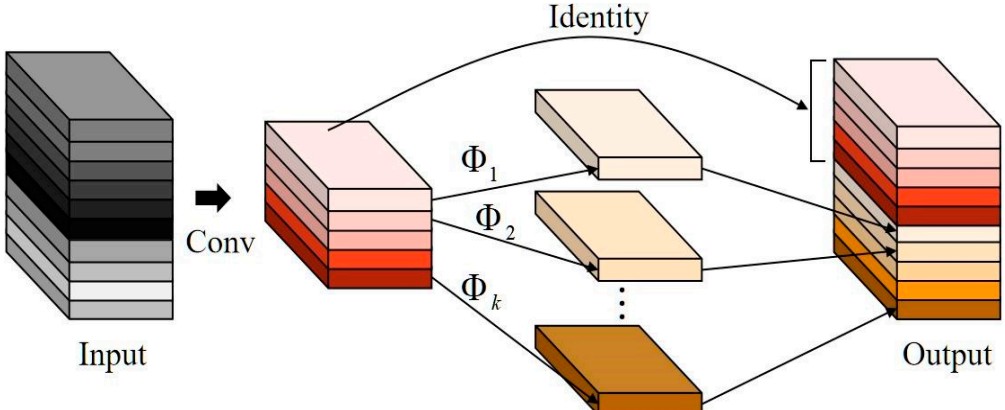

**Figure 6.** Ghost module structure diagram.

As shown in Figure 6, the Ghost module first performs conventional convolution operations on the input image to obtain the feature map of half the channel number of the input image. Then, it performs deep convolution on the obtained feature map to generate the result of the corresponding channel number. In the next step, the Concat operation is performed between the profound convolution result and the original feature map, and the number of channels in the output image is expanded to the same as that in the input image.

The Ghost module's effect is equivalent to a conventional convolution, but it dramatically reduces the computation through this particular structure. Therefore, many researchers have improved their lightweight models by using the design idea of Ghost modules instead of conventional convolution.

### 2.3. Forest Fire Dataset

Although there are countless forest fires worldwide every year, gathering images of fires during the fire period is challenging. To study the fire test, 8000 images of open fire and fire smoke were selected to showcase the characteristics of forest fires. Among them, 6500 were used as the training set and 1500 as the verification set (Figure 7).

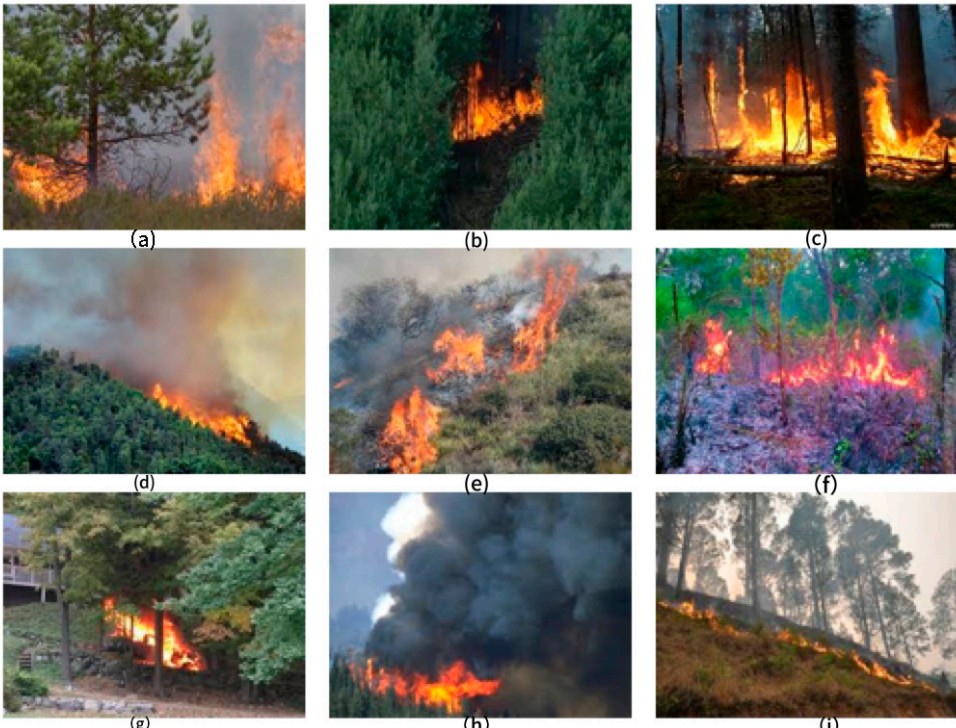

**Figure 7.** Forest fire detection dataset. (**a–i**) are the nine datasets we selected and we gave it a number for visualisation and analysis in Section 2.4.2, the original model misses the regions with red arrows in (**e,i**). The YOLOv5n-CB model proposed in this paper identifies the regions that are not detected by the YOLOv5n model.

### 2.4. Analysis of the Multi-Stage Improvement Results of YOLOv5

2.4.1. Test Environment and Results Analysis

The performance verification test of this algorithm is carried out on the Windows 10 operating system with the above self-made forest fire dataset as the dataset, in which 6500 pictures are used as the training set and 1500 pictures as the verification set, including two types of targets: fire and smoke. In the experiment, the Pytorch deep learning framework is used for model training and reasoning, CUDA is installed to support GPU side training, and the three attention fusion mechanism models of YOLOv5 + SE, YOLOv5 + ECA, YOLOv5 + CBAM, and the BiFPN structure fusion model are compared with the original model. The results are shown in Table 1.

**Table 1.** Comparison of performances.

| Model | Precision (%) | Recall (%) | Parameter (PCS) | mAP@0.5 (%) | GFLOPs |
|---|---|---|---|---|---|
| YOLOv5n | 87.8 | 85.7 | 1,766,623 | 89 | 4.2 |
| YOLOv5n + SE | 88.2 | 86 | 1,774,815 | 89.4 | 4.2 |
| YOLOv5n + ECA | 86.9 | 86.6 | 1,766,626 | 89.5 | 4.2 |
| YOLOv5n + CBAM | 88.1 | 87.6 | 1,774,913 | 89.7 | 4.2 |
| YOLOv5n + BiFPN | 87.5 | 87.1 | 1,783,007 | 89.6 | 4.3 |

As seen from Table 1, among the three attention mechanisms, the integration of the CBAM attention module has the most apparent improvement on model performance but exerts little impact on the number of model parameters and the amount of computation. Therefore, this paper finally chooses the integrated CBAM attention module as an improved target model attention fusion mechanism method. At the same time, it is shown that the missing rate of the YOLOv5n model in combination with the BiFPN structure is reduced, effectively improving the model's performance, and it is therefore considered an effective optimization strategy.

### 2.4.2. Performance Analysis of YOLOv5n-CB Algorithm after Multi-Stage Improvement

Through experiments with the above-discussed improved YOLOv5n network at different stages, it is verified that the improved methods have various degrees of improvement targets for the model performance. Therefore, an enhanced strategy combining the CBAM attention mechanism and the BiFPN structure was added to the original YOLOv5n model and named YOLOv5n-CB. The new model was then trained, and the P-R curve comparison between the improved model and the original model and the mAP value index comparison figure are shown in Figures 8 and 9.

According to the comparison diagram of the P-R curve and mAP value index, it can be seen that the performance of the proposed YOLOv5n-CB model is better. The specific training indexes are shown in Table 2.

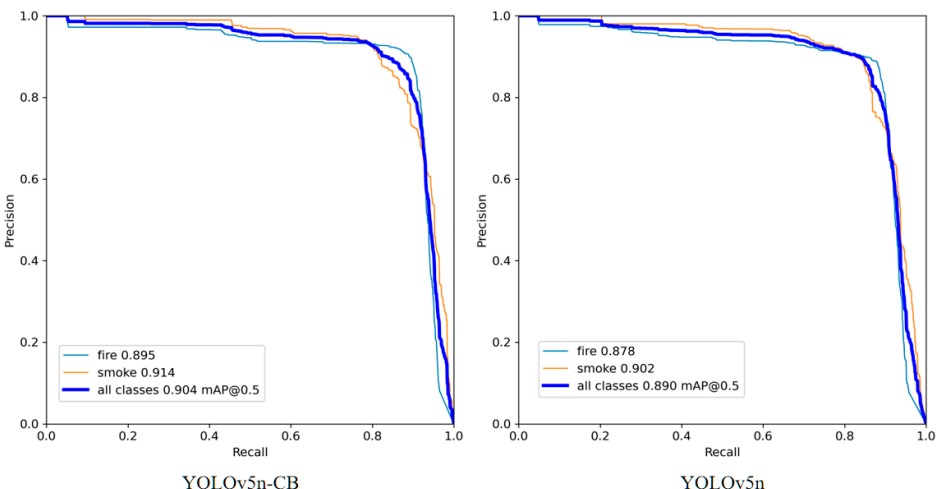

**Figure 8.** Comparison of network's P-R curve diagram.

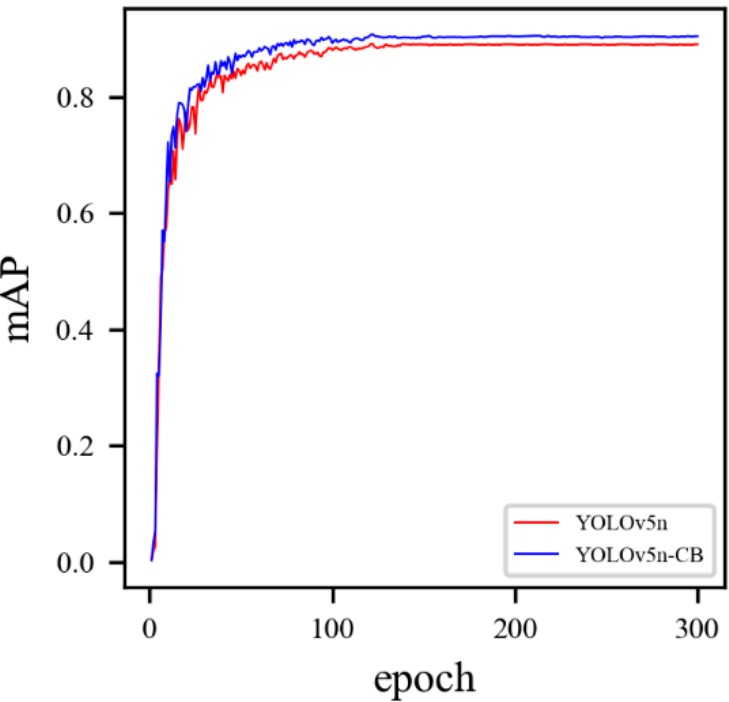

**Figure 9.** Comparison of network's mAP metrics diagram.

**Table 2.** YOLOv5n-CB network performance.

| Model | Precision (%) | Recall (%) | Parameter (PCS) | mAP@0.5 (%) | GFLOPs |
|---|---|---|---|---|---|
| YOLOv5n | 87.8 | 85.7 | 1766623 | 89 | 4.2 |
| YOLOv5n-CB | 88.6 | 87.7 | 1791297 | 90.4 | 4.3 |

In Table 2, the accuracy rate and recall rate of the YOLOv5n-CB model are significantly improved compared with the original model, increased by 0.8% and 2%, respectively. This shows that the model's false and missing detection rates for the verification set detection target are reduced. Still, the number of participants only increases by 1.7%, which does not significantly increase the model size. The mAP value index is increased by 1.4%, and the model's performance is improved. This verifies the effectiveness of the proposed method.

The experiments have shown that the performance of the YOLOv5n-CB algorithm model proposed in this paper is better. The results of model recognition are visualized and analyzed using the test pictures; part of the recognition effect is shown in Figure 10. The red arrows refer to the flame targets missed by the original model (e) and (i) in Figure 7.

As can be seen from Figure 6, the YOLOv5N-CB model can identify the categories of fire and smoke presented in the figure, while the YOLOv5n model fails to detect them. At the same time, the YOLOv5n-CB model not only identifies the missed flame target of the original model but also has a higher confidence score than the original model, indicating that the model identification is more accurate. Furthermore, the prediction frame is close to the actual frame, indicating that the improved model has a good detection effect, proving the proposed model's application value.

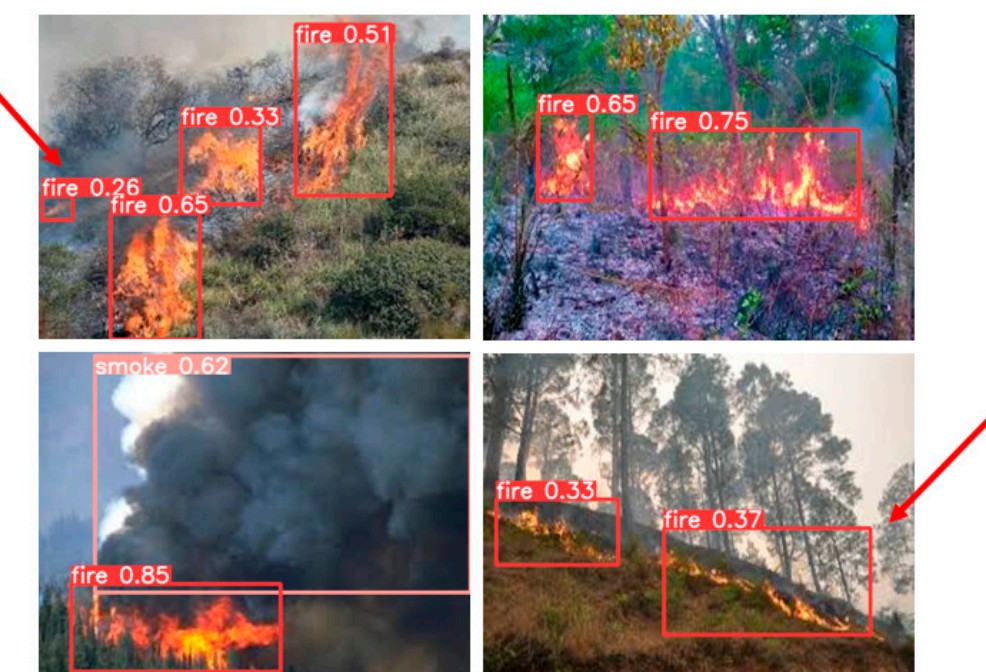

**Figure 10.** YOLOv5n-CB recognition results diagram.

## 3. Improve the Lightweight and Model Deployment of the YOLOv5 Network

In practical application scenarios, low-performance object detection algorithms are often deployed on terminals. Therefore, it is necessary to improve the lightweight model to compensate for the terminal computing power deficiency. In this regard, it is planned to carry out a variety of lightweight improvements on the above proposed improved algorithm and analyze the performance of the improved model to select the optimal lightweight improvement strategy and deploy the improved lightweight model on the Jetson Nano device to test its running speed and detection effect.

### 3.1. Performance Analysis of Improved G-YOLOv5-CB Network Model

According to the above experimental verification, applying the lightweight strategy of the GhostNet network to the YOLOv5n-CB model is necessary. The new model is named G-YOLOv5-CB, whose structure is shown in Figure 11.

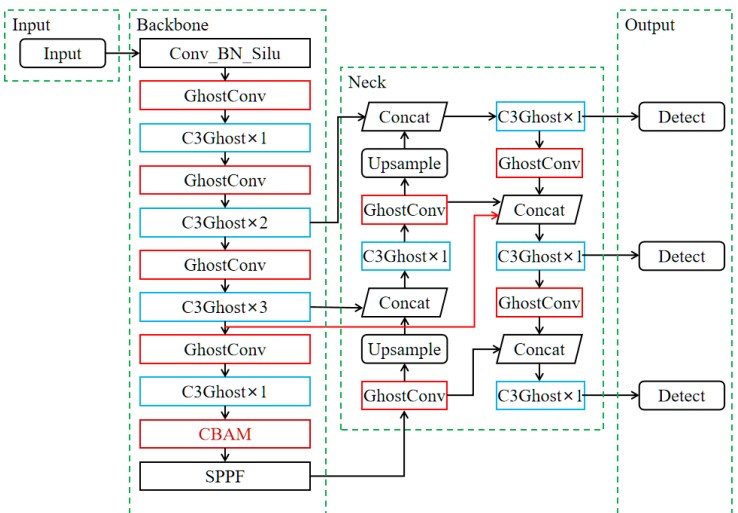

**Figure 11.** G-YOLOv5-CB model structure diagram.

The G-YOLOv5-CB model was trained with the same training configuration as above. The P-R curve of the training process of the model is shown in Figure 12, where the AP values of fire and smoke targets, namely, the area under the curve, are both about 0.9, which proves that the proposed improved G-YOLOv5-CB model has excellent performance in feature extraction and feature fusion of targets.

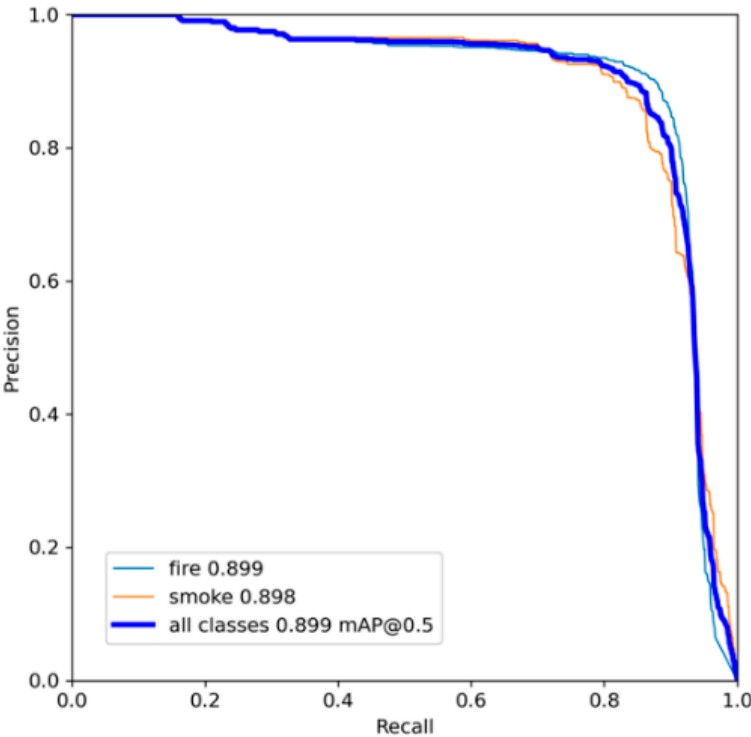

**Figure 12.** Trained G-YOLOv5n-CB network P-R curve diagram.

The proposed model is compared with other deep learning detection models, and the results are shown in Table 3.

**Table 3.** Model comparison results.

| Model | Parameter | Weight File Size (MB) | mAP@0.5 (%) |
|---|---|---|---|
| YOLOv5n | $17.66 \times 10^6$ | 3.65 | 89 |
| YOLOv3-tiny | $21.73 \times 10^6$ | 16.7 | 79.4 |
| YOLOv5s | $70.25 \times 10^6$ | 13.7 | 90.6 |
| Faster R-CNN | $28.32 \times 10^6$ | 108.4 | 85.2 |
| G-YOLOv5-CB | $9.68 \times 10^6$ | 2.2 | 89.9 |

The experiments show that the complexity of the G-YOLOv5-CB model is significantly reduced compared with other models for better real-time performance at the expense of a minor precision loss. Therefore, the lightweight model is more suitable for deployment on embedded devices.

*3.2. Jetson Nano Model Deployment and Testing*

The forest fire real-time monitoring system designed in this paper adopts Jetson Nano as the detection platform for development. After setting up the environment, we copy the YOLOv5 file package containing the changes in the G-YOLOv5n-CB model relative to the original model and the best-pt file saved with the training results of the above model, respectively, into the Jetson Nano system and the project, and use best-pt as the detection model to capture the input image. After completing the deployment of the model, parts of the images from the forest fire dataset are selected for testing on the Jetson Nano

development board, and the specific test results are shown in Figure 13; the model can detect both the fire and smoke targets presented in the picture. The average detection time is 0.065 s and the average frame rate is about 15 FPS. The fast detection speed of the model proves the effectiveness of the lightweight strategy and can meet the real-time requirements of forest fire detection.

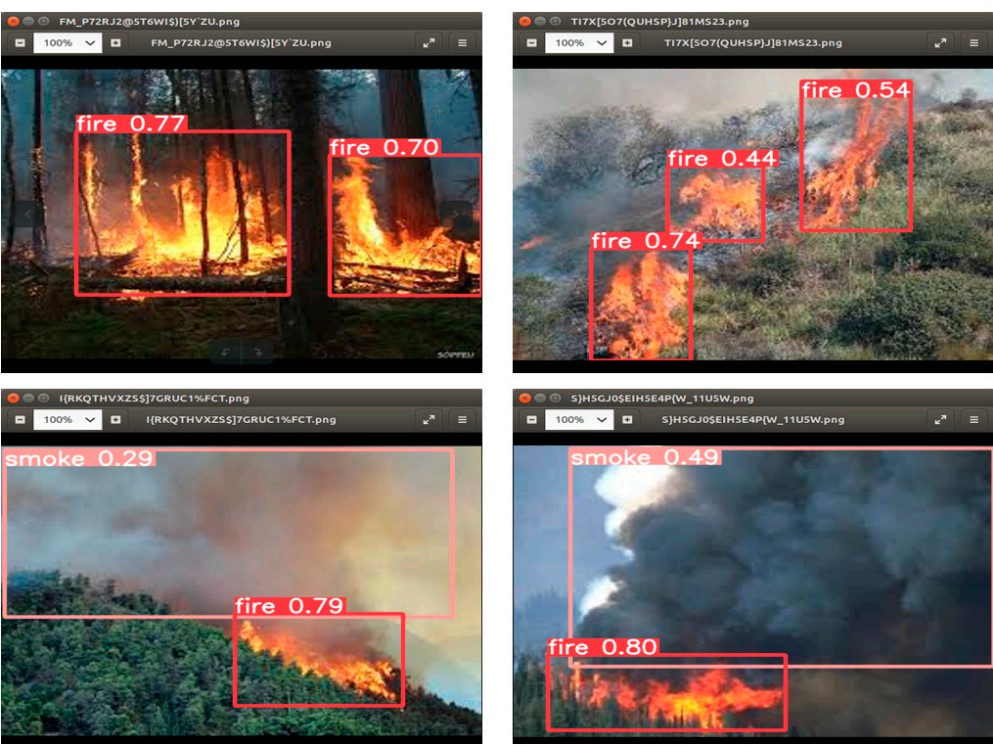

**Figure 13.** Jetson Nano test results.

## 4. Design of Campus Forest Fire Real-Time Monitoring System

The overall design of this system comprises three components: forest fire detection module, carrier platform automatic navigation module, and upper computer display and control module [14], as shown in Figure 14.

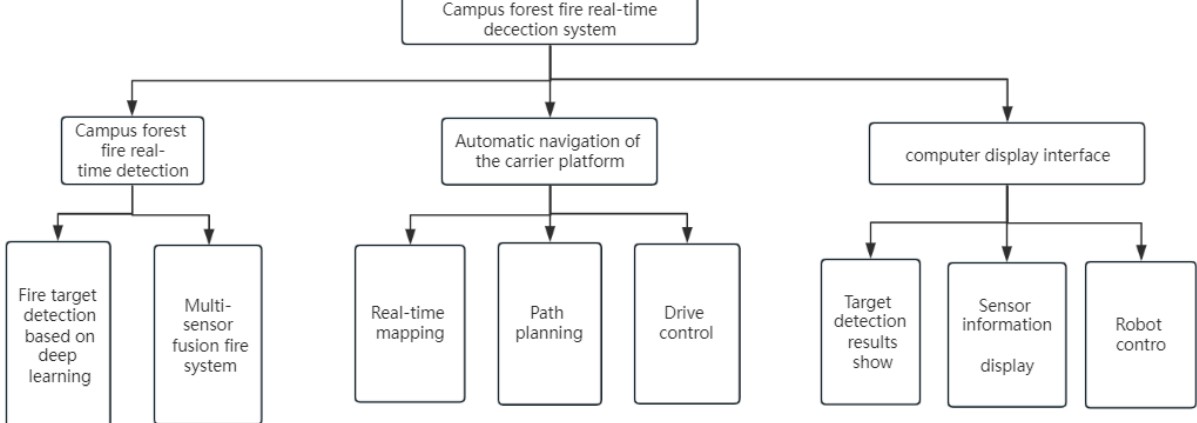

**Figure 14.** Overall system design diagram.

To fulfill the field operation in the campus woodland environment, this study adopts an intelligent vehicle as a carrier platform to carry out a forest fire detection function module, cruise in the campus woodland, and conduct real-time detection. A smart vehicle terminal's function mainly includes a campus forest fire detection module and a map navigation path planning module. The hardware connection diagram of the carrier platform is shown in Figure 15.

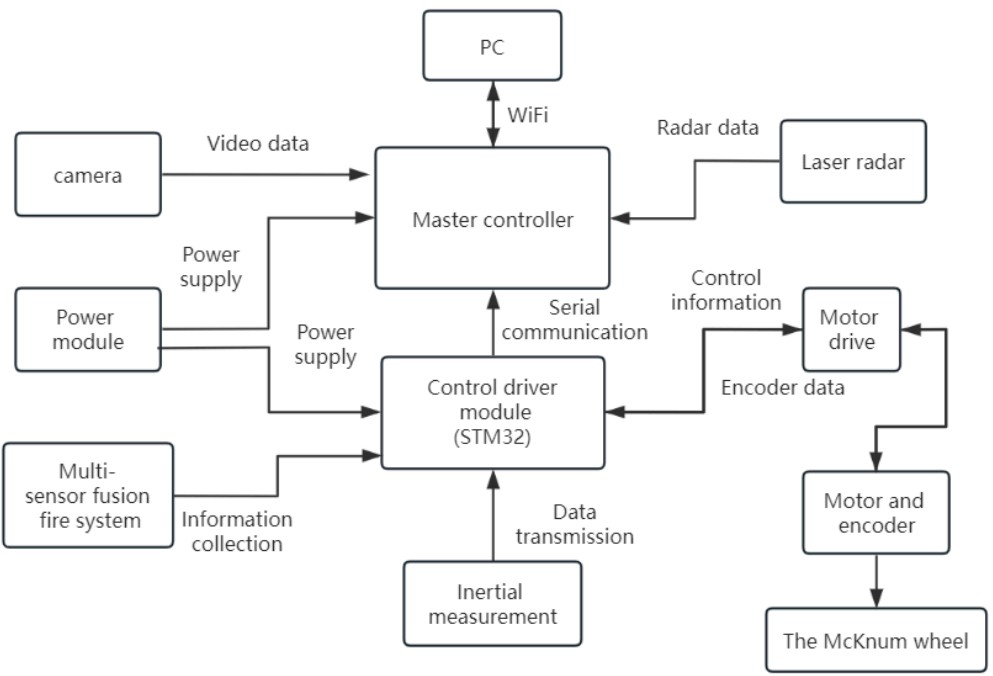

**Figure 15.** System hardware connection diagram.

*4.1. Design of Campus Forest Fire Detection Module*

In the design of the environmental awareness module, the sensors used in the carrier platform mainly include the encoder at the tail of the motor, 2D LiDAR, and RGB-D depth camera. In this system, the AB phase incremental Hall encoder in the photoelectric encoder is used to obtain the motor information through the S2L radar and LeEco's Astra depth camera, which can receive the steering state of the robot while measuring the speed. The multi-sensor forest fire detection module supplements the forest fire target detection method [15], composed of a CO gas sensor, flame sensor, and smoke sensor. When two or more sensors reach the set alarm threshold conditions, it will send alarm information to the system. The process is shown in Figure 16.

Here, an MQ-2 smoke sensor, an MQ-7 CO gas sensor, and a flame sensor are all capable of measuring the conductivity of the gas sensor material to determine the concentration of the measured gas in the air. Its detection performance at standard temperature and pressure has long-term validity, and the work is stable and reliable. The flame sensor detects the flame target in the environment through the sensitivity of the flame spectrum and the detection distance changes with the size of the flame. The larger the flame and the farther the detection distance, the better the detection effect.

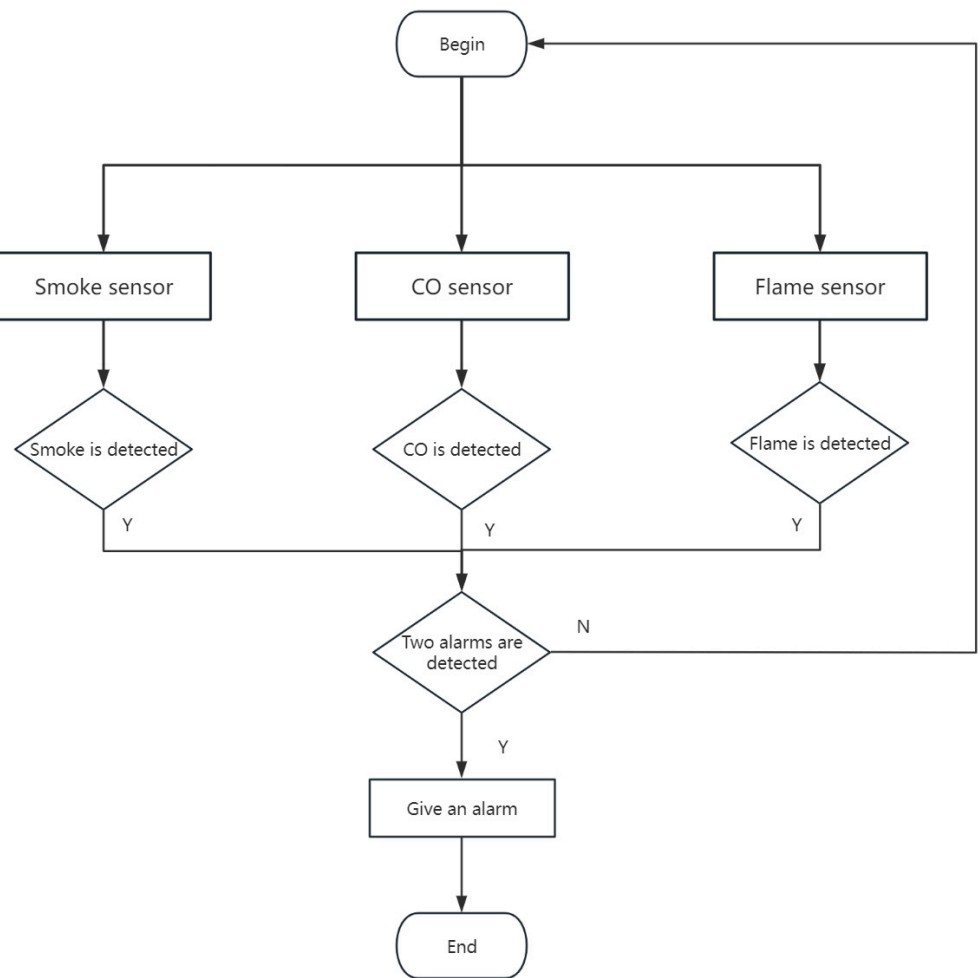

**Figure 16.** Multi-sensor fire detection module.

*4.2. Automatic Navigation System Design*

4.2.1. Overall Design of Automatic Navigation Function Module

The robot's autonomous navigation in the campus woodland undergoes the process from the set starting point to the destination, and how it navigates is similar to autonomous driving. The structure of the automatic navigation system is shown in Figure 17.

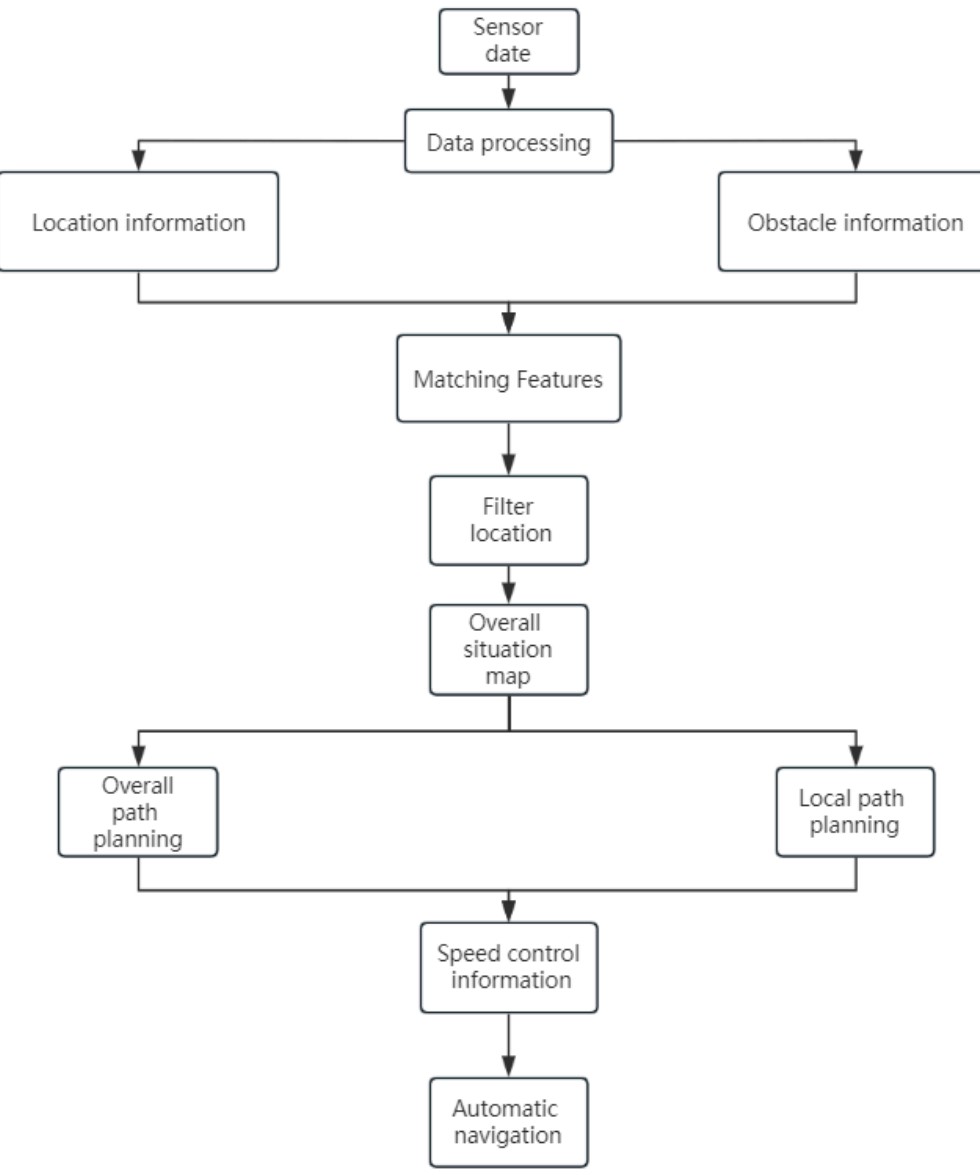

**Figure 17.** Automatic navigation module diagram.

4.2.2. Navigation Map Construction

The Cartographer algorithm based on the graph optimization theory proposed by the Google team is adopted as the map construction algorithm in this study [16]. It mainly includes sensor data processing, local mapping, and global optimization [17]. The overall framework flow of the algorithm is shown in Figure 18.

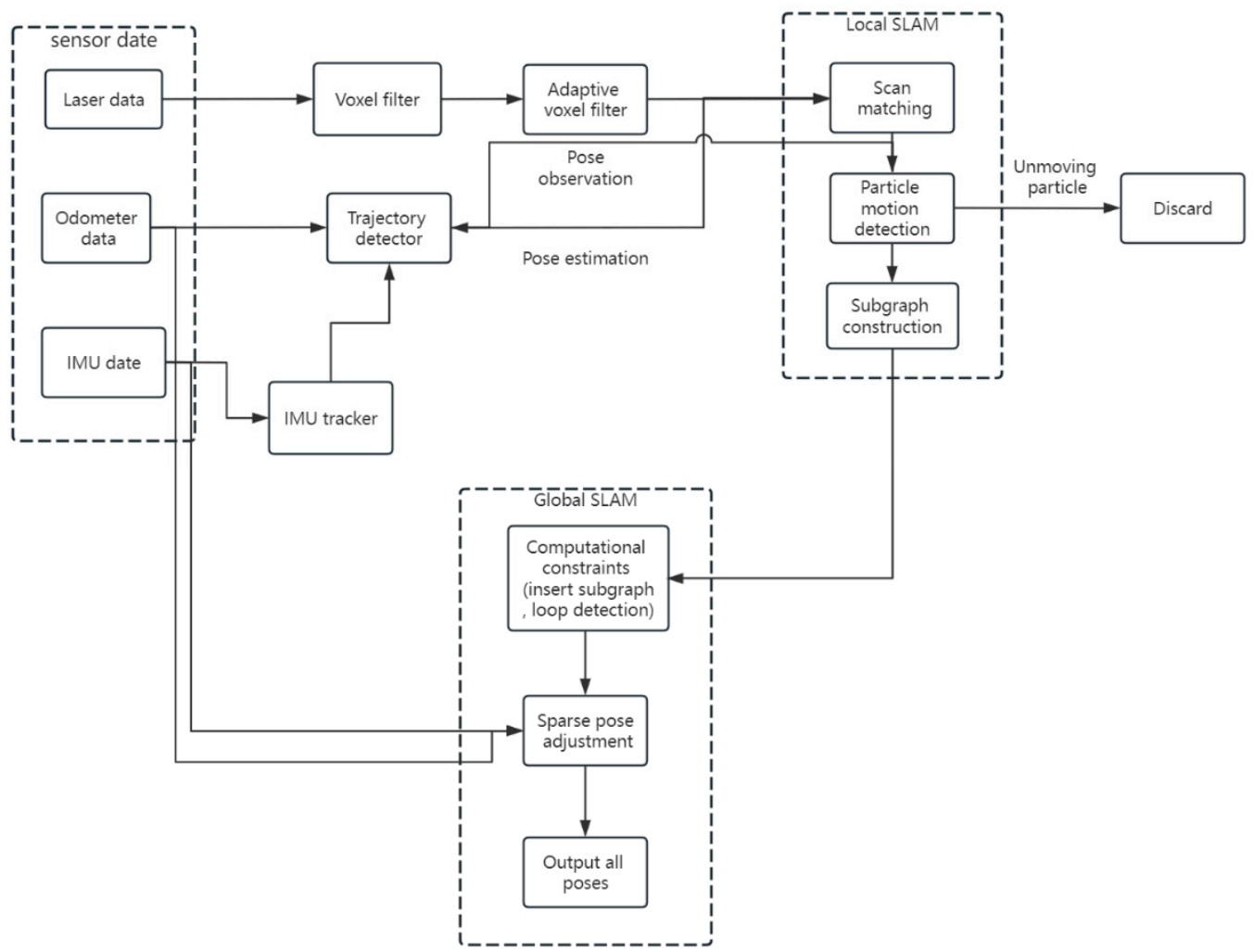

**Figure 18.** Cartographer algorithm framework.

### 4.2.3. Map Construction Experiment

To verify the mapping capability of the Cartographer algorithm in actual scenarios, this experiment uses the Cartographer algorithm in the ROS system to construct raster maps that are convenient for the subsequent movement of robots. To control the robot more conveniently during the process of drawing construction, the robot is generally controlled by a keyboard to move in the drawing environment. After the keyboard control function is opened by inputting the keyboard control command on the PC side, the robot can be controlled to realize the walking and turning functions in the natural environment. The Rivz interface in Figure 19 shows the real-time mapping scene when the robot walks through the whole area in the experiment scene. The white area in the figure is the map being built, and the green lines are the map boundaries scanned by the radar during operation.

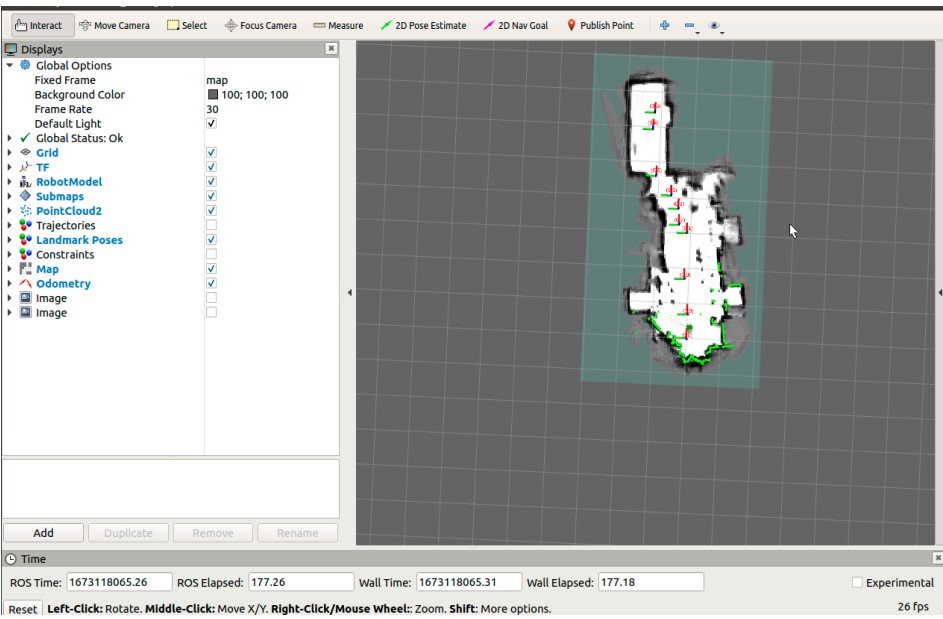

**Figure 19.** Simulate the robot map construction process.

After the map is built, the saved raster map is shown in Figure 20. After comparing actual scenarios, it can be concluded that the maps created by the Cartographer algorithm have high reducibility to the natural environment and less noise, thus laying a good foundation for implementing navigation functions.

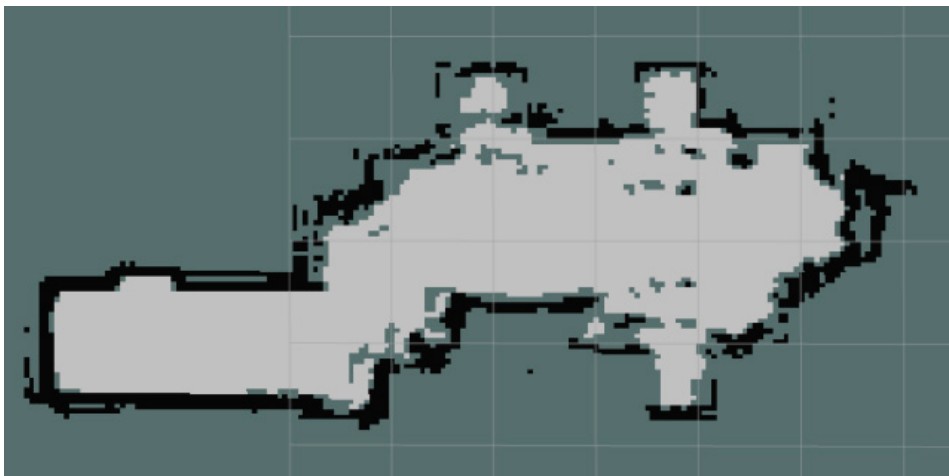

**Figure 20.** Raster map construction renderings.

### 4.2.4. Path Planning

After the establishment of the map, to achieve the robot navigation function, the robot's driving route needs to be planned, mainly consisting of global and local path planning [18].

In global path planning, the A* algorithm is selected to carry out the robot's global path planning [19]. This method is more targeted for the target point in the design, so the search range of the route is narrowed, and the speed of path planning is extensively promoted. The algorithm flow is shown in Figure 21.

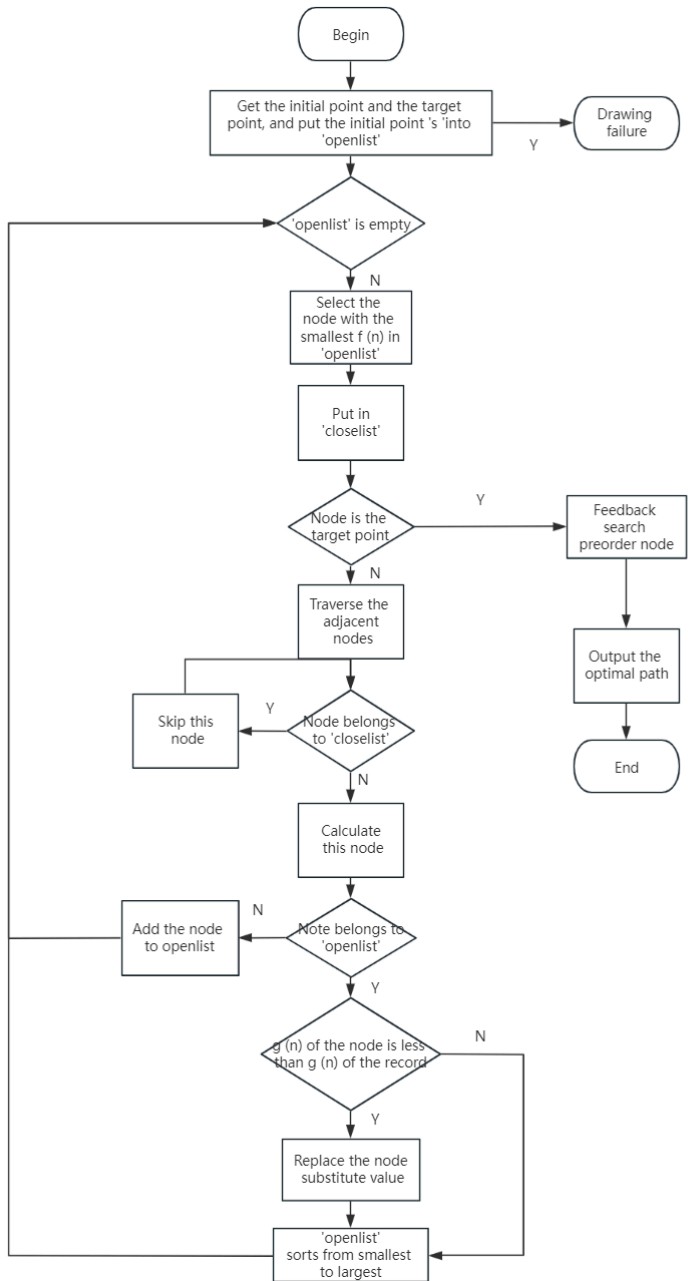

**Figure 21.** A* algorithm flow chart.

The A* algorithm searches the map path by estimating the moving cost of nodes on the map, and its calculation formula is shown in Formula (1).

$$Fn = Gn + Hn \tag{1}$$

In Formula (1), $G_n$ is the generation value of the generated path from the starting point to the specified node. $H_n$ is the heuristic method used by the algorithm to estimate the generation value of moving from the selected node to the endpoint. $F_n$ is a global estimate of generational value from start to finish.

For the A* algorithm, the $H_n$ calculation involves many factors, including obstacle information, path direction, etc. In this paper, the Euclidean distance, the linear distance

method, is used to analyze the robot's movement, the calculation formula of which is shown in Equation (2).

$$Hn = \sqrt{(Xn - Xgoal)^2 + (Yn - Ygoal)^2} \tag{2}$$

In Formula (2), the coordinate of the robot's current position is $(X_n, Y_n)$, the coordinate of the target position is $(X_{goal}, Y_{goal})$, and Hn is the Euclidean distance between the two points.

### 4.2.5. Local Path Planning

The algorithm flow is shown in Figure 22.

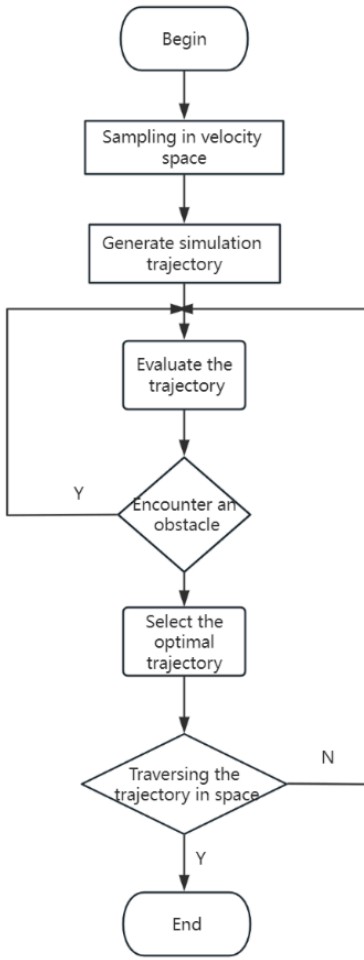

**Figure 22.** DWA algorithm flow chart.

In Figure 21, the DWA algorithm first samples the velocity space of the robot and then estimates the possible moving trajectory of the robot in the velocity space. The evaluation function evaluates each trajectory. If there is an obstacle, the trajectory is skipped; if there is no obstacle, the optimal trajectory is selected. The robot's status is finally updated after traversing all generated trajectories in space.

### 4.2.6. AMCL Positioning Algorithm

The AMCL algorithm, also called the adaptive Monte Carlo positioning algorithm, is a positioning algorithm based on a particle filter [20]. This algorithm uses random virtual particles to estimate the motion state of the robot in the process of driving and updates. It evaluates the motion state of the robot in real time through alternate resampling and KLD sampling methods to achieve accurate robot positioning.

### 4.2.7. Navigation Test

After a study of related positioning algorithms and navigation algorithms, the base map constructed by the Cartographer algorithm is used to test and analyze the navigation functions of target points during the robot's actual work. First, the saved two-dimensional map is loaded into the ROS system to realize navigation functions, and the AMCL function package is used to locate the robot. Then, the move base function package is used to carry out path planning and movement control for the robot, and the global path planning algorithm A* and local path planning algorithm DWA under the move base function package are invoked to assist the robot in completing navigation. The robot navigation process is shown in Figure 23.

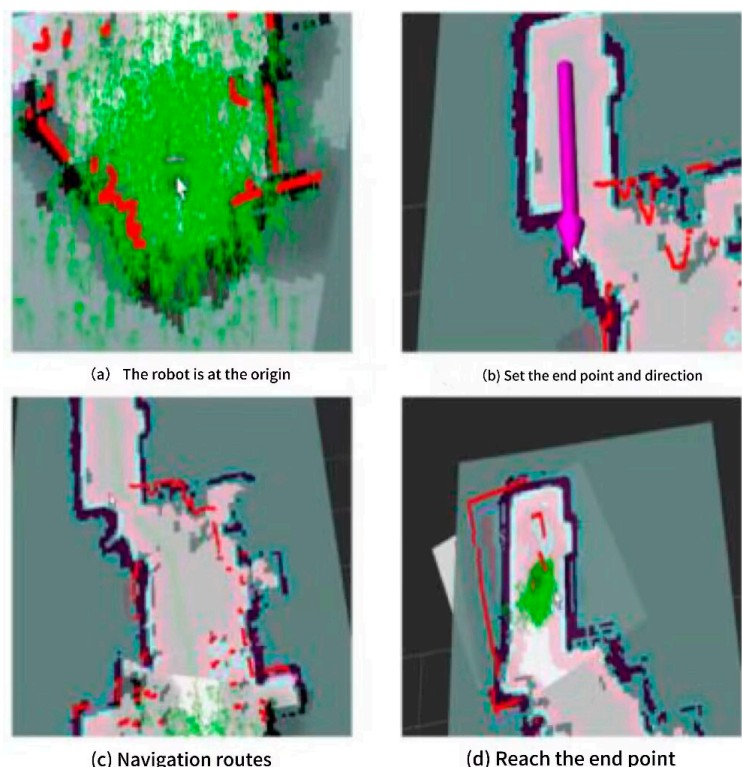

**Figure 23.** Robot navigation process.

Figure 23a shows the starting point position of the robot. Figure 23b shows the setting of the endpoint position and direction in the Rviz interface using the 2D Nav Goal tool mouse. The green line in Figure 23c is the global path planned by the A* algorithm, and the short red line segment in front of the robot is the local path generated by the DWA algorithm. Figure 23d shows the robot reaching the endpoint. The experiment demonstrates that the robot can reach the endpoint following the established route and realize free navigation between the two points in the process of global navigation. At the same time, the navigation obstacle avoidance function is tested by placing obstacles on the robot's route.

## 5. Test of Campus Forest Fire Real-Time Monitoring System

### 5.1. Upper Computer Interface Design

The upper computer interface is based on the PyQt5 design of the campus forest fire real-time monitoring system interface. Using a GUI programming method to write a visual interface, PyQt5 is a kind of human–computer interaction software that can be operated through the peripherals without inputting instruction codes. The PyQt5 interface can be designed to display the detection results more intuitively. Different buttons display target detection results, robot control, sensor information, and fire alarm information. Among them, the implementation process of the forest fire detection module of the system is first to

collect environmental information through the USB camera and then to conduct real-time detection on the Jetson Nano embedded platform. At the same time, the detection result is transmitted to the display interface of the upper computer in real time through Wi-Fi and socket protocol. The specific execution flow chart is shown in Figure 24.

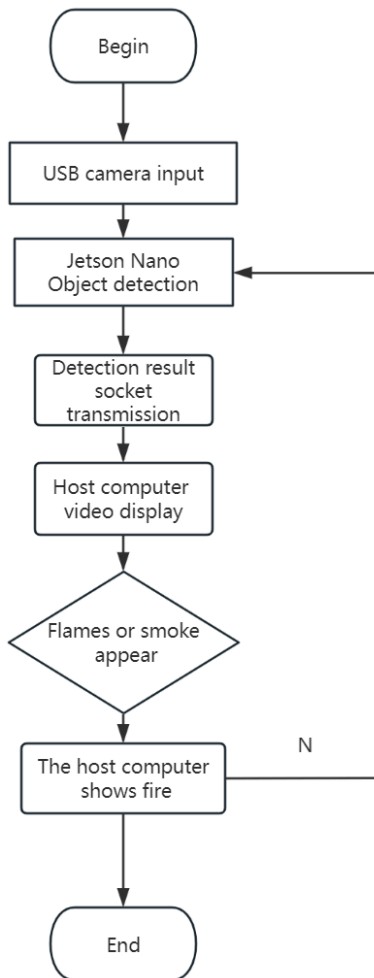

**Figure 24.** Forest fire detection system flow chart.

*5.2. Automatic Navigation Function Test in Campus Woodland Scene*

As shown in Figure 22, the outdoor laser SLAM construction of the carrier platform can reflect the natural outdoor environment. After the map construction is completed, click 2D NAV GOAL in the RVIZ tool to set the location of the target point and test the avoidance function of the obstacle test system by simulating the pedestrian in the forest scene as the obstacle.

Figure 25a shows the robot at the starting point, with the simulation of the pedestrians existing in the actual woodland scene as obstacles. Figure 25b shows that the system successfully identifies the obstacles. Figure 25c shows that the system successfully avoids obstacles in the navigation process. The green line in the figure is the global planned path, and the short red line is the local planned path. Figure 25d shows that the system successfully reaches the target point and realizes the campus woodland scene's navigation and obstacle avoidance functions.

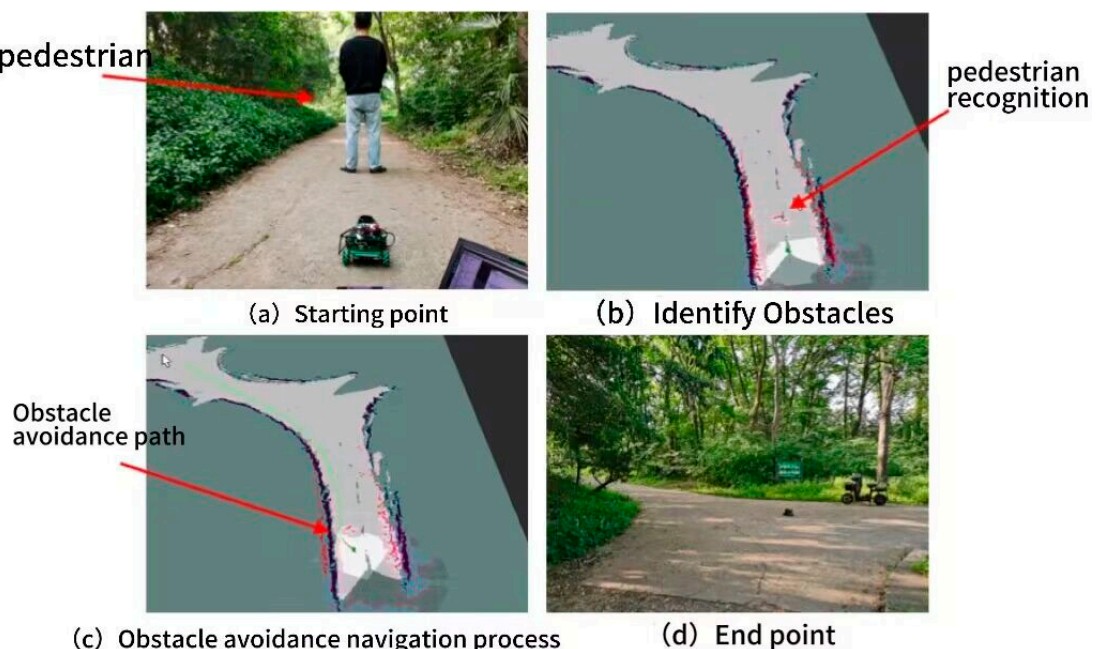

**Figure 25.** Navigational obstacle avoidance test process in campus woodland scene.

## 6. Conclusions

1. A forest fire dataset is constructed. The forest fire image data are obtained through the Internet, and the images are annotated by the Make Sense tool. A total of 8000 pictures containing fire and smoke are made to convey the fire information, and the forest fire dataset for the test is completed.

2. A YOLOv5n-CB algorithm with higher detection accuracy is proposed. Based on the YOLOv5n model, the CBAM attention mechanism is introduced at the junction of the backbone and neck to improve network performance effectively while introducing only a small number of parameters. The idea of a weighted bidirectional feature pyramid network (BiFPN) is introduced to enhance the neck part and strengthen the feature extraction of the target. Experiments have shown that the YOLOv5n-CB model integrated with the CBAM attention mechanism and BiFPN structure has a higher accuracy and recall rate than the original model, the mAP value index on the forest fire dataset has increased by 1.4%, and the network performance has been significantly improved.

3. Aiming at the problem of large volume and complex calculation of the algorithm model, the G-YOLOv5n-CB algorithm is proposed, and it can be deployed on the Jetson Nano platform for real-time forest fire detection. The MobileNetV3 network and GhostNet network are used to improve the lightweight feature extraction network of the original algorithm. The test shows that integrating the lightweight strategy of GhostNet reduces the number of parameters and the amount of computation and keeps the drop in detection accuracy of the mAP value index at only 0.5%. By comparing it with other in-depth models, the effectiveness of the proposed algorithm is verified.

4. The G-YOLOv5n-CB model is deployed on the Jetson Nano platform, and the operating environment of the model is configured. According to the experiment, the detection speed of the model reaches 15 FPS, which meets the real-time requirements of the system.

5. The global path planning algorithm A* and local path planning algorithm DWA are used in the automatic navigation system of the carrier platform of this system to plan the path of the robot, which realizes the target point navigation and obstacle avoidance functions of the robot in the campus woodland scene.

6. Based on PyQt5, the upper computer interface of the system is developed to realize real-time display of detection results and action control of the robot. The comprehensive test shows that the system can carry out the real-time monitoring of campus forest fires accurately.

**Author Contributions:** The real-time fire monitoring system in the campus forest land was developed by D.X.; Q.W. proposed the G-YOLOv5n-CB forest fire algorithm; the verification experiment and the writing of the article were carried out by J.C.; and the overall scheme was designed and the results and analysis conducted by Z.W. All authors have read and agreed to the published version of the manuscript.

**Funding:** This research is sponsored by the 2023 Jiangsu Higher Education Teaching Reform Research Project (2023JSJG716).

**Data Availability Statement:** The data supporting this study's findings are available upon request from the corresponding author.

**Acknowledgments:** Many thanks to Wang Jun of the Nanjing Forestry University for his assistance in sample processing. And the authors appreciate Wang Xiwei for proofreading and language editing of the final revision.

**Conflicts of Interest:** The authors declare no conflicts of interest.

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
