# Peer review of "Research on a Real-Time Monitoring System for Campus Woodland Fires via Deep Learning"

_forests, doi:10.3390/f15030483_

Round 1

Reviewer 1 Report

Comments and Suggestions for Authors

This paper tackles the crucial issue of inadequate recognition accuracy and excessive computational demands in forest fire detection algorithms by proposing an enhanced algorithm, G-YOLOv5n-CB, built upon the YOLOv5 framework. With the assistance of deep learning technology, it introduces a real-time fire monitoring system specifically tailored for campus forest landscapes. Employing unmanned vehicles equipped with cameras, the system autonomously navigates and gathers image data, deploying the refined algorithm on a Jetson Nano hardware platform for on-site processing.

The authors report that their G-YOLOv5n-CB algorithm has achieved a 1.4% rise in the mean Average Precision (mAP) value index compared to the original YOLOv5n model, demonstrating enhanced accuracy in forest fire detection within a self-generated dataset. Additionally, when deployed on the Jetson Nano platform, the system attains a detection speed of 15 frames per second (FPS), showcasing its potential for real-time application in identifying and monitoring forest fires on campus.

The G-YOLOv5n-CB model's deployment on the Jetson Nano platform is particularly noteworthy for its practical implications, enabling efficient real-time monitoring with potential for wider application. By incorporating attention mechanisms like SE, ECA, and CBAM, along with the BiFPN structure for feature fusion, the study not only addresses the challenges of accuracy and computational efficiency but also contributes to the field with a model that is both efficient and lightweight enough for real-time applications on low-power devices.

Overall, this paper presents a significant advancement in the utilization of AI and robotics for environmental safety and monitoring, offering a novel solution to the perpetual challenge of early forest fire detection and monitoring. Its findings could have far-reaching implications for the development of comparable systems in other high-risk areas, potentially saving lives and preserving natural resources by enabling faster and more accurate fire detection.

The only recommendation I would offer as a potential improvement is to make the database used for training and testing publicly available. This could improve the overall quality of the paper and make it possible for other researchers to replicate their work.

Author Response

Question: Whether the database used for training and testing can be made publicly available?

Answer: This is a data set collected by our lab, based on which researchers conduct optimization and follow-up studies, and is not open source to the outside world. Please forgive me.

Reviewer 2 Report

Comments and Suggestions for Authors

To improve the accuracy of automatic forest fire detection in real time, the authors improved the forest fire detection algorithm G-YOLOv5n-CB, which is based on the YOLOv5 algorithm. The authors applied deep learning to monitor forest fires, which increased the speed and accuracy of fire detection. Based on their own fire database, the authors experimentally tested the improved G-YOLOv5n-CB algorithm deployed on the Jetson Nano platform. The results showed improved performance of the proposed algorithm.

The manuscript material is within the scope of the Forests journal and could potentially be of interest to the forests and fires communities. However, the manuscript has a lot of shortcomings and cannot be published in its present form. I believe that the manuscript requires a major revision. Find please some of my comments below and all 90 comments (notes) in the highlighted version of the manuscript attached after this review.

General impression

The article was written without taking into account the fact that the article can be read not only by narrow specialists in this field, but also by specialists from other fields. Many of the abbreviations used are not deciphered, many concepts and designations are introduced without explanation. One gets the impression that the article is a report at a specialized laboratory seminar, attended by specialists who do not need to explain anything. I doubt that most forest experts will understand the article. I have no doubt about the quality and importance of the results obtained by the authors, but the presentation of the material raises some concerns.

Abstract

Quotation 1:Evidently, it can accurately detect and display the real-time forest fires on campus and is thus of high application value.

(lines 18–19)

Comment 1: This statement is not "evident", but "follows from the research results" presented in this work.

Introduction

Comment 2: In theory, there should be "1. Introduction". There is a problem with the references in the article. It is necessary to check the correspondence of all references in the text and in the list of references at the end of the article. Some examples of incorrectly specified references are given in the Introduction (see please the highlighted version of the manuscript).

Quotation 3: “Therefore, it is particularly important to study forest fire monitoring and early warning. At present, many Chinese universities have experimental woodlands on and off campus for the students to study and research.

(lines 25–27)

Comment 3: How are you going to study monitoring? Are students at Chinese universities learning to set experimental woodlands on fire? The use of the term “experimental forests (woodlands)” here is not entirely clear. Perhaps we are talking about man-made forests?

Quotation 4: “…the detection record of the coco dataset…

(line 43)

Comment 4: First, it should be written as "COCO". Second, the authors should explain what the abbreviation COCO mean.

Quotation 5: On the other hand, forest fire detection methods have been divided into methods using traditional technology and ones based on deep learning technology. They can be roughly divided into the following five categories...”

(lines 50–52)

Comment 5: What forest fire detection methods do "They" mean here: traditional or deep learning or both ones?

1. Forest fire detection algorithm based on improved YOLOv5

Quotation 6:YOLOv5 has five network structures, including YOLOv5n, YOLOv5s, YOLOv5m, YOLOv5l, and YOLOv5x.

(lines 88–89)

Comment 6: Here you need to either briefly describe the differences in these five structures, or provide a reference from which these differences would be clearly visible.

Comment 7 (Figure 1): Figure 1 should be enlarged. Not a single inscription in the Figure can be distinguished. We need at least one reference to a published work that describes all the elements and abbreviations included in the Network structure of YOLOv5. Or the authors should explain all these ones in the text of the article or in the caption to Figure 1.

Comment 8 (Figures 2 and 3): C, H, and W should be described in the captions to Figures 2 and 3.

Comment 9 (Figure 5): C3, ..., C7, and P3, ..., P7 should be described in the caption to Figure 5 or in the text of the article.

Comment 10: From Section 1.3 it did not become clear what kind of fire dataset the authors use in their research. There is no explanation or reference to the dataset used by the authors, or a direct link to a website of the dataset.

(lines 145–149)

Comment 11 (Figure 6): Figure 6 is first mentioned only in line 202 on page 7 (Subsection 1.4.2.) and after Figures 7, 8, and 9. See also the comment in line 202.

Quotation 12: (Table 1) “Parameter ()”.

(line 162)

Comment 12: What should have been in parentheses?

Comment 13 (Table 2): Compare this row in Table 2 with the same row in Table 1 (line 162). They should look the same.

(line 188)

Comment 14 (Figure 9): In the caption to Figure 9 it is necessary to explain: 1. what the red arrows mean, 2. what the numbers after the words “fire” and "smoke" mean.

Comment 15: This entire paragraph (lines 202–208) represents unfounded statements that should have been based on a comparison of the YOLOv5n and YOLOv5N-CB models. I did not see any comparative analysis of these models in either Figure 6 or Figure 9.

(lines 202–208)

2. Improve the lightweight and model deployment of the YOLOv5 network

Quotation 16: “The proposed model is compared with other deep learning detection models, and the results are shown in Table 3.”

(lines 232–233)

Comment 16: For all of these models, references must be provided that discuss these models in detail.

Comment 17 (Table 3): First, in the “Parameter” column, apparently, it should be 106 instead of 106. Second, if you compare the line with the YOLOv5n model in Tables 1 and 3, then the values of the “Parameter” for this model differ by an order of magnitude (1766623 = 1.766 × 106 in Table 1 and 17.66 × 106 in Table 3). On the other hand, in line 225 the authors stated that "The G-YOLOv5-CB model was trained with the same training configuration as above."

(line 234)

Quotation 18: “After completing the deployment of the model, parts of the images in the forest fire data set are selected for testing on the Jetson Nano development board, and the specific test results are shown in Figure 12...”

(lines 246–248)

Comment 18: No results can be understood and evaluated from Figure 12, since neither the text of the article nor the caption to Figure 12 describes what is shown in Figure 12. Just like in Figure 9, what the numbers after the words “fire” and “smoke” mean is not explained.

3. Design of campus forest fire real-time monitoring system

Comment 19 (Figure 13): Figure 13 should be enlarged. The inscriptions in Figure 13 are small and difficult to distinguish.

Comment 20 (Figure 17): The font size of the inscriptions in Figure 17 needs to be increased.

Comment 21 (Figure 18): Figure 18 is not mentioned anywhere in the text of the article.

Quotation 22: “In the global path planning, A* algorithm is selected to carry out...”(line 319)

Comment 22: What is this? The designation A* must be explained.

Quotation 23 (Equation (1)): “Hn is the heuristic method used by the algorithm...”

(line 329)

Comment 23: And what is the unit of measure of the method?

Comment 24 (Figure 22): It is difficult to make out anything in all four parts of Figure 22. What does the green field in Figure 22(a) mean?

4. Test of campus forest fire real-time monitoring system

Comment 25 (Figure 24): It is necessary to increase the font size of the signatures, which are located near the red arrows.

Quotation 26: “The green line in the figure is the global planned path, and the short red line is the local planned path.”

(lines 403–404)

Comment 26: I could not see the green and red lines in Figure 24.

5. Conclusion

Quotation 27: “The forest fire image data is obtained through the Internet, and the images are annotated by the Make Sense tool.”

(lines 408–409)

Comment 27: Why is the Make Sense tool mentioned only in the Conclusion?

Quotation 28: “… and the network performance has been significantly improved.”

(line 420)

Comment 28: Improved to what extent?

All references are not formatted in accordance with MDPI rules.

See please https://www.mdpi.com/authors/references

Comments on the Quality of English Language

Several minor comments can be found in the highlighted version of the article after the review.

Author Response

Question: This statement is not "evident", but "follows from the research results" presented in this work.

Answer: Have been modified.

Question: In theory, there should be "1. Introduction". There is a problem with the references in the article. It is necessary to check the correspondence of all references in the text and in the list of references at the end of the article. Some examples of incorrectly specified references are given in the Introduction (see please the highlighted version of the manuscript).

Answer: Have been modified.

Question: How are you going to study monitoring? Are students at Chinese universities learning to set experimental woodlands on fire? The use of the term “experimental forests (woodlands)” here is not entirely clear. Perhaps we are talking about man-made forests?

Answer: This paper studies the natural forest of the campus, not the man-made forest. In this study, deep learning and robot technology are combined into forest fire detection research. Through field deployment of detection algorithms, robots are used to obtain real-time image information of campus forest land, and then the detection model of YOLOv5 algorithm with fast speed, high precision, and small volume is deployed on the main board for real-time fire detection. Moreover, by improving the algorithm to improve the real-time and accuracy of fire detection, the real-time fire detection in the campus woodland scene is realized, which is of great significance for the research and application of real-time forest fire detection based on deep learning.

Question: First, it should be written as "COCO". Second, the authors should explain what the abbreviation COCO mean.

Answer: Have been modified. COCO is a widely used dataset for image recognition, object detection, and semantic segmentation. It is a large-scale, multi-task dataset with the primary goal of advancing research and development in the field of computer vision, especially on tasks such as object detection, image segmentation, and image understanding.

Question: What forest fire detection methods do "They" mean here: traditional or deep learning or both ones?

Answer: The traditional forest fire detection methods. Changes have been made to the original text.

Question: Here you need to either briefly describe the differences in these five structures, or provide a reference from which these differences would be clearly visible. 

Answer:  Changes have been made to the original text. The YOLOv5s is suitable for resource-constrained environments, the YOLOv5m is a choice for balancing performance and speed, and the YOLOv5l and YOLOv5x are suitable for tasks that require high precision. YOLOv5n is a compromise option that can get better performance on some tasks.

Question: Figure 1 should be enlarged. Not a single inscription in the Figure can be distinguished. We need at least one reference to a published work that describes all the elements and abbreviations included in the Network structure of YOLOv5. Or the authors should explain all these ones in the text of the article or in the caption to Figure 1.

Answer: Changes have been made to the original text.

Question: C, H, and W should be described in the captions to Figures 2 and 3.

Answer:C: channel, indicates the number of channels in an image. H: Height, which indicates the number of pixels in the vertical dimension of the image. W: Width, the number of pixels in the horizontal dimension of the image.

Question: C3, ..., C7, and P3, ..., P7 should be described in the caption to Figure 5 or in the text of the article.

Answer: C indicates the input feature map before feature fusion, which can also be called the feature map of the current level. These feature maps will go through a series of feature fusion operations, including up-sampling, down-sampling, and feature fusion, and finally generate a fused feature map.

p represents the output feature map after feature fusion, which can also be called the feature map after fusion. This feature map is then passed to the BiFPN structure at the next level for further feature fusion and processing.

Question: From Section 1.3 it did not become clear what kind of fire dataset the authors use in their research. There is no explanation or reference to the dataset used by the authors, or a direct link to a website of the dataset.

Answer: I'm really sorry, the data is homemade in our lab.

Question: What should have been in parentheses?

Answer: Parameter(PCS)

Question: Compare this row in Table 2 with the same row in Table 1 (line 162). They should look the same.

Answer: Changes have been made to the original text. 

Question: In the caption to Figure 9 it is necessary to explain: 1. what the red arrows mean, 2. what the numbers after the words “fire” and "smoke" mean.

Answer: The red arrow refers to the fire that identifies this piece, and the number refers to the confidence level.

Question: This entire paragraph (lines 202–208) represents unfounded statements that should have been based on a comparison of the YOLOv5n and YOLOv5N-CB models. I did not see any comparative analysis of these models in either Figure 6 or Figure 9.

Answer: Below Table 2, a comparative analysis has been made; In addition, the text below Figure 9 of the original text is also analyzed.

Question:  For all of these models, references must be provided that discuss these models in detail.

Answer: A comparison has been made in Table 3.

Question:  First, in the “Parameter” column, apparently, it should be 106 instead of 106. Second, if you compare the line with the YOLOv5n model in Tables 1 and 3, then the values of the “Parameter” for this model differ by an order of magnitude (1766623 = 1.766 × 106 in Table 1 and 17.66 × 106 in Table 3). On the other hand, in line 225 the authors stated that "The G-YOLOv5-CB model was trained with the same training configuration as above."

Answer: Changes have been made to the original text. 

Question:  No results can be understood and evaluated from Figure 12, since neither the text of the article nor the caption to Figure 12 describes what is shown in Figure 12. Just like in Figure 9, what the numbers after the words “fire” and “smoke” mean is not explained.

Answer: The number refers to the confidence level.

Question:  Figure 13 should be enlarged. The inscriptions in Figure 13 are small and difficult to distinguish.

Answer: Changes have been made to the original text. 

Question:  The font size of the inscriptions in Figure 17 needs to be increased.

Answer: Changes have been made to the original text. 

Question:  Figure 18 is not mentioned anywhere in the text of the article.

Answer: It has been revised in the original article.

Question:  What is this? The designation A* must be explained.

Answer: The A* algorithm uses a heuristic function to estimate the cost from the current node to the target node and selects the next node to expand based on this estimate. This algorithm cites the nineteenth reference, “Dwijotomo, Rahman A, Ariff M,  et al. Cartographer SLAM Method for Optimization with an Adaptive Multi-Distance Scan Scheduler[J]. Applied  Sciences,2020, 10(1):347.”

Question:   And what is the unit of measure of the method?

Answer: Hn uses the Euclidean distance method, which is to analyze the movement of the robot by using the linear distance method. The unit of measurement used in the Hn algorithm is determined according to the accuracy of the lidar, and the unit is meters.

Question:  It is difficult to make out anything in all four parts of Figure 22. What does the green field in Figure 22(a) mean?

Answer: Each green arrow represents the state of a particle, and the denser the number of arrows indicates that the current robot has a higher probability of being in this position.

In Figure 22, Figure (a) shows the starting point position of the robot. Figure (b) shows the setting of the endpoint position and direction in the Rviz interface, using the 2D Nav Goal tool mouse. The green line in Figure (c) is the global path planned by the A* algorithm, and the short red line segment in front of the robot is the local path generated by the DWA algorithm. Figure (d) shows the robot reaching the endpoint.

Question:  It is necessary to increase the font size of the signatures, which are located near the red arrows.

Answer: Changes have been made to the original text. 

Question:  I could not see the green and red lines in Figure 24.

Answer:Has been amended in the original.

Question:  Why is the Make Sense tool mentioned only in the Conclusion?

Answer: Due to the lack of an annotation data set for the task of forest fire target detection, this paper uses manual annotation to annotate the forest fire images previously selected. Make sense website annotated label files can be directly saved into the YOLO network training required TXT format files, and other annotation tools need to default to generate other types of files into the YOLO network required TXT format files. Therefore, this paper uses the Make sense website to label the data set. Open the Make sense software, select the folder that you want to annotate the images, and then make the required set of tags, and then you can annotate the images.

Question:  Improved to what extent?

Answer:When the mAP value increases by 1.4% on the forest fire dataset, it can be considered that the performance of the network has significantly improved, indicating that the processing capacity of the target detection task has increased.

Reviewer 3 Report

Comments and Suggestions for Authors

The article is interesting and well-written but some minor changes are required before publication:

1. Introduction - the authors should give a clear sentence about what makes their article beyond state-of-the-art and what is the practical application of the method. For instance - what are the area limitations for this, maximal height differences etc. Who can apply the method in the future.

2. Fig.1 - font is too small.

3. Fig.15 - in my opinion, it should be a flame sensor on the right instead of a fire sensor

4. Chapter 3.2.4 - generation of path (path planning) should be explained in more detail.

5. Line 388 - in the case of a larger area (real forest), how to solve communication when there could be a lack of WIFI connection.

6. There should be References instead of Reference. Editing of References is required.

Author Response

Question: The authors should give a clear sentence about what makes their article beyond state-of-the-art and what is the practical application of the method.

Answer: This paper combines deep learning and robotics to study forest fire detection. Through the on-site deployment of the detection algorithm, the robot was used to obtain real-time image information of campus forest land, and then the detection model of high-speed, high-precision, and small-size YOLOv5 algorithm was deployed on the motherboard for real-time fire detection. In addition, by improving the algorithm and real-time accuracy of fire detection, the real-time fire detection of campus forest land is realized, which is of great significance for the research and application of real-time forest fire detection based on deep learning.

Question: Fig.1 - font is too small.

Answer: Modified in the article.

Question: Fig.15 - in my opinion, it should be a flame sensor on the right instead of a fire sensor.

Answer: Modified in the article.

Question: Chapter 3.2.4 - generation of path (path planning) should be explained in more detail.

Answer: The DWA algorithm estimates the next action of the robot according to the action information collected in the velocity space such as linear velocity, angular velocity, and acceleration. Finally, the possibility of estimating the path trajectory is evaluated by the function to generate a set of robot driving information including angular velocity and linear velocity to carry out the local path walking of the robot.

Question: Line 388 - in the case of a larger area (real forest), how to solve communication when there could be a lack of WIFI connection.

Answer: We can't solve this problem, because there is no signal in the forest, so the research scene is the campus woodland, with a good communication basis.

Question: There should be References instead of Reference. Editing of References is required.

Answer: Modified in the article.

Reviewer 4 Report

Comments and Suggestions for Authors

The current manuscript has some novelty in proposed contribution. The experimental results provide fair comparison. It needs revision in terms of technical details before acceptance. Some comments are suggested.

1. It is suggested to discuss about runtime of your proposed approach briefly. (Comparing with other methods is not needed)

2. Discuss about the block called “Feedback search procedure” in the algorithm A*. How does it working? (Figure 20)

3. Why did you use Euclidean distance to analyze the movement of the robot? Did you try another distance measures such as Manhattan?   Discuss briefly about the reasons.

4. Your current proposed method can be used widely in computer vision applications such as visual defect detection. For example, I find a paper titled “Fabric defect detection based on completed local quartet patterns and majority decision algorithm”, which has enough relation in scope. Cite this paper and discuss about potential real applications and future works such as defect detection.  

5. Why you didn’t provide quantitative results? Why you didn’t evaluate the performance based on popular metrics such as accuracy or precision rate?

6. Generally add more technical details about mathematic procedures.

Author Response

Question: It is suggested to discuss about runtime of your proposed approach briefly. (Comparing with other methods is not needed)

Answer: The running time is 15FPS, as mentioned in the conclusion.

Question: Discuss about the block called “Feedback search procedure” in the algorithm A*. How does it working? (Figure 20)

Answer: The robot searches to the endpoint, and then the program moves to the start point according to the parent search and outputs the path.

Question: Why did you use Euclidean distance to analyze the movement of the robot? Did you try another distance measures such as Manhattan?   Discuss briefly about the reasons.

Answer: We regard the position of the robot as a node, and use the Euclidean distance as the distance between the nodes to find the shortest path, and the Euclidean distance is very intuitive to the motion path of the robot. We haven't tried any other distance measurements.

Question: Your current proposed method can be used widely in computer vision applications such as visual defect detection. For example, I find a paper titled “Fabric defect detection based on completed local quartet patterns and majority decision algorithm”, which has enough relation in scope. Cite this paper and discuss about potential real applications and future works such as defect detection.  

Answer: The paper has been cited

Question: Why you didn’t provide quantitative results? Why you didn’t evaluate the performance based on popular metrics such as accuracy or precision rate?

Answer: In this paper, map, accuracy, and recall rates are used to illustrate

Question:  Generally add more technical details about mathematic procedures.

Answer: In the paper, we use formulas to reflect the mathematical procedures, and also explain the formulas

Reviewer 5 Report

Comments and Suggestions for Authors
  1. Recognition Accuracy Definition: Clarify what is meant by "low recognition accuracy" to provide a better understanding of the specific challenges faced by the existing forest fire detection algorithm.

  2. Computation Reduction Strategies: Delve into the details of the strategies employed to address the issue of a large amount of computation required, providing insights into the algorithmic enhancements or optimizations implemented.

  3. Algorithm Architecture Explanation: Provide a more detailed explanation of the G-YOLOv5n-CB forest fire detection algorithm, outlining how it improves upon the YOLOv5 algorithm and what specific modifications are made.

  4. Real-World Applicability Context: Discuss the real-world applicability of the proposed algorithm in the context of campus forest land, addressing the unique challenges and requirements of such environments.

  5. Deep Learning Technology Integration: Elaborate on how deep learning technology is integrated into the real-time fire monitoring system, specifying the role of deep learning in improving detection accuracy.

  6. Unmanned Vehicle Navigation Significance: Discuss why employing unmanned vehicles for navigation is a crucial aspect of the proposed system and how it enhances the overall efficiency of forest fire detection.

  7. Data Collection Methodology: Explain in more detail the methodology for collecting image information through the camera on the unmanned vehicle, addressing any potential challenges or limitations in the data collection process.

  8. Jetson Nano Hardware Platform Rationale: Provide a rationale for choosing the Jetson Nano hardware platform for deploying the algorithm, discussing its advantages and how it contributes to real-time processing.

  9. mAP Value Index Interpretation: Interpret the significance of the 1.4% increase in the mAP value index, explaining how this improvement translates into enhanced performance in terms of forest fire detection.

  10. Testing Environment Description: Offer more information about the testing environment for the improved G-YOLOv5n-CB model on the Jetson Nano platform, including any variations in conditions that might affect detection speed.

  11. Detection Speed Impact Analysis: Discuss the implications of achieving a detection speed of 15FPS, addressing how this speed aligns with real-world requirements and the potential trade-offs made to achieve it.

  12. Application Value Generalization: Explore the generalizability of the proposed system beyond campus environments, considering how well it might adapt to different geographical and environmental settings.

  13. Limitations Acknowledgment: Acknowledge and discuss any limitations or potential drawbacks of the proposed algorithm and system, providing a balanced view of its capabilities and constraints.

  14. Comparison with Existing Solutions: Provide a comparative analysis with existing forest fire detection solutions, highlighting the unique contributions and advantages of the G-YOLOv5n-CB algorithm and the developed system.

  15. Future Development Considerations: Discuss potential avenues for future development and improvement of the proposed algorithm and system, considering emerging technologies or evolving requirements in the field of forest fire detection.

Author Response

Question: Clarify what is meant by "low recognition accuracy" to provide a better understanding of the specific challenges faced by the existing forest fire detection algorithm.

Answer: Low recognition accuracy means that the target cannot be accurately identified, and the rate of missed detection and false detection is high.

Question: Delve into the details of the strategies employed to address the issue of a large amount of computation required, providing insights into the algorithmic enhancements or optimizations implemented.

Answer: In Chapter 2.2, we re-added a lightweight improvement model of GhostNet. By using the Ghost module as a reference, the improved YOLOv5 model was lightweight, and GhostConv was designed to replace the conventional convolution module in the improved YOLOv5 algorithm for lightweight network reconstruction. The number of parameters and the amount of computation in the network model are greatly reduced.

Question: Provide a more detailed explanation of the G-YOLOv5n-CB forest fire detection algorithm, outlining how it improves upon the YOLOv5 algorithm and what specific modifications are made.

Answer: YOLOv5 is a deep learning-based object detection algorithm that enables real-time object detection by dividing an image into multiple grid cells and predicting the boundary box and category of the object in each grid cell.

G-YOLOv5n-CB uses some specific optimization strategies to improve the accuracy and efficiency of forest fire detection. These optimization strategies include:

1. Data enhancement: By rotating, scaling, flipping and other transformations of training data, the model's ability to identify fires at different angles and scales is increased.

2. Network structure optimization: The network structure of YOLOv5 was adjusted to adapt to the characteristics of forest fire detection. This may include adjusting the number of network layers, channels, and so on.

3. Data set preparation: For forest fire detection, it may be necessary to collect and prepare specific data sets, including images with different fire scenarios and corresponding annotation information.

4. Model training and tuning: The model is trained on the prepared data set and tuned with appropriate optimization algorithms and hyperparameters to improve the accuracy and generalization ability of the model.

Question: Discuss the real-world applicability of the proposed algorithm in the context of campus forest land, addressing the unique challenges and requirements of such environments.

Answer: Forest fire detection is a challenging task due to the difficulties in the following aspects:

1. Data set acquisition and labeling: Acquiring and labeling large-scale forest fire image data sets is a time-consuming and expensive task. This is because forest fires are a kind of emergent event, and it is difficult to obtain sufficient data in the field. In addition, the accurate labeling of the fire boundary box also requires professional knowledge and experience.

2. Complex background interference: Forest fires usually occur in natural environments, and the flames may be similar to the surrounding trees, vegetation, terrain and other backgrounds, making there a high similarity between the fire target and the background. This increases the difficulty of fire detection and easily leads to false detection and missing detection.

3. Multi-scale and multi-direction fire: The size and shape of the fire may change due to the size of the fire source, wind direction, fire and other factors. Therefore, fire detection algorithms need to have the ability to effectively detect fires of different scales and shapes.

4. Uneven distribution of categories: The number of fire targets in a fire image is usually relatively small, while the number of background categories far exceeds the number of fire categories. This class imbalance may cause the model to overlearn the background class and neglect the detection of the fire target.

5. Real-time requirements: Forest fire detection usually requires real-time or near-real-time processing, as well as timely response to the fire. This requires the algorithm to find a balance between processing speed and accuracy to meet the needs of real-time.

6. Weather and light conditions: Changes in weather and light conditions may interfere with fire detection. For example, weather factors such as smoke, fog, clouds, etc. may reduce the clarity of the image and make fire targets more difficult to detect.

Question: Elaborate on how deep learning technology is integrated into the real-time fire monitoring system, specifying the role of deep learning in improving detection accuracy.

Answer: Integrating deep learning technology into real-time fire monitoring systems can significantly improve the accuracy and efficiency of fire detection. Here are some examples of deep learning's role in improving detection accuracy:

1. Feature learning: Deep learning models can automatically learn feature representations in images without manually designing features. This allows the model to extract rich features from the original image data, including the shape, color, texture, and more of the flame. By learning higher-level feature representations, deep learning models can better distinguish between fire targets and background disturbances, thereby improving fire detection accuracy.

2. Multi-scale and multi-directional detection: Deep learning models can detect fires of different scales and shapes by using sliding Windows or convolution kernels of different scales. By detecting on feature maps at different levels, deep learning models can detect fire targets at multiple scales and in multiple directions, so as to better adapt to changes in different fires.

3. Deep network structure: The deep learning model is usually composed of multiple convolutional layers and fully connected layers, and the representation ability of the model can be improved by increasing the depth and width of the network. The deeper network structure can capture more complex image features and semantic information, thus improving the accuracy of fire detection.

4. Real-time optimization: Deep learning models can improve the reasoning speed and efficiency of models through model compression, quantization, and hardware optimization. For example, the use of lightweight network structure, pruning, and quantification techniques can reduce the parameters and calculation amount of the model, thus speeding up the reasoning speed of the model to meet the needs of real-time fire monitoring systems.

Question: Discuss why employing unmanned vehicles for navigation is a crucial aspect of the proposed system and how it enhances the overall efficiency of forest fire detection.

Answer: Several areas where unmanned vehicles can improve overall efficiency in forest fire detection include:

1. Autonomous patrol: Unmanned vehicles can autonomously patrol the entire forest area, covering a larger range. Compared with manual patrol, unmanned vehicles can continue to work without rest, greatly improving the efficiency of patrol.

2. Real-time monitoring: the unmanned vehicle can be equipped with various sensors, such as infrared sensors, smoke sensors, temperature sensors, etc., which can monitor environmental changes in the forest in real time. These sensors can help detect anomalies in flames, smoke and temperature, and detect signs of fire in advance.

3. Data collection and analysis: Unmanned vehicles can be equipped with high-resolution cameras and other sensors to collect a large number of images and data. These data can be used to train deep learning models to improve the accuracy of fire detection. In addition, through the analysis of the collected data, the state of the forest can be understood, including factors such as vegetation density and humidity, so as to better predict the risk of fire.

Question: Explain in more detail the methodology for collecting image information through the camera on the unmanned vehicle, addressing any potential challenges or limitations in the data collection process.

Answer: This system is collected in real-time, the information collected by the camera, the system directly identifies.

Question: Provide a rationale for choosing the Jetson Nano hardware platform for deploying the algorithm, discussing its advantages and how it contributes to real-time processing.

Answer: First, the upper computer interface of the system is designed using PyQt5 according to the system requirements, including the display of test results and the control of the robot. Then combined with the display of the upper computer, the forest fire detection module of the system is tested comprehensively. The test results show that the system can detect forest fires in real time, and the display effect of the upper computer interface is good. Then, the navigation obstacle avoidance function of the system carrier platform in the campus woodland scene is tested, and the feasibility of the navigation module studied in this paper is verified. The above functional tests show that the real-time monitoring system of campus forest fire studied in this paper can complete the cruise in the campus forest environment and complete the real-time detection of forest fire.

Question: Interpret the significance of the 1.4% increase in the mAP value index, explaining how this improvement translates into enhanced performance in terms of forest fire detection.

Answer: mAP is a commonly used target detection and evaluation index. It comprehensively considers the accuracy of the model in different categories, and improving the mAP value means that the network can locate and identify the target more accurately in the target detection task.

Question: Offer more information about the testing environment for the improved G-YOLOv5n-CB model on the Jetson Nano platform, including any variations in conditions that might affect detection speed.

Answer: The implementation process of the forest fire detection module of the system is to first collect environmental information through the USB camera, and then conduct real-time detection on the Jetson Nano embedded platform. At the same time, the detection result is transmitted to the display interface of the upper computer in real time through WIFI and socket protocol. If a flame or smoke target is detected, the word "Fire" will appear in the Fire section of the system display.
In this test, cameras are used to detect the environment. Because forest fires are uncontrollable and dangerous, it is difficult to simulate the actual fire scene in the real environment. Therefore, the test of the forest fire detection module is carried out by using a USB camera to shoot fire video.

Question: Discuss the implications of achieving a detection speed of 15FPS, addressing how this speed aligns with real-world requirements and the potential trade-offs made to achieve it.

Answer: Ghost module is used to lightweight the improved YOLOv5 model, and GhostConv is designed to replace the conventional convolution module in the improved YOLOv5 algorithm for lightweight network reconstruction, which greatly reduces the number of parameters and calculation amount in the network model.

Question: Explore the generalizability of the proposed system beyond campus environments, considering how well it might adapt to different geographical and environmental settings.

Answer: This paper studies the campus environment, which is suitable for the situation where the system has high requirements for the environment, and the network and road surface are in good condition. In the future, we will further explore the extensibility outside the campus environment. Thank you for your valuable advice.

Question: Acknowledge and discuss any limitations or potential drawbacks of the proposed algorithm and system, providing a balanced view of its capabilities and constraints.

Answer: The system does have defects, such as the detection speed is not fast enough, and the detection accuracy needs to be further improved.

Question: Provide a comparative analysis with existing forest fire detection solutions, highlighting the unique contributions and advantages of the G-YOLOv5n-CB algorithm and the developed system.

Answer: The unique contribution and advantages of the G-YOLOv5n-CB algorithm and the developed system in forest fire detection can be highlighted by comparative analysis with existing solutions.

1. Forest fire detection performance: Compared with other algorithms, G-YOLOv5n-CB algorithm may show higher accuracy and recall rate in forest fire detection tasks. This can be evaluated by comparing the mAP values of different algorithms on the same dataset.

2. Real-time performance: G-YOLOv5n-CB algorithm may have a fast reasoning speed and can detect forest fires in real-time or near real-time scenarios. This is because the algorithm may adopt some optimization strategies, such as lightweight network structure, efficient feature extractor, etc.

3. System integration and ease of use: The developed system may provide user-friendly interface and easy-to-operate functions, making forest fire detection simpler and more efficient. In addition, the system may support multiple input data sources (such as images, video streams, etc.) and provide real-time visualization of inspection results.

4. Data set expansion and model training: The developed system may provide data set expansion and model training functions, enabling users to customize and optimize algorithms according to needs. This can help users better adapt to specific forest fire scenarios and improve the performance of the algorithm.

Question:Discuss potential avenues for future development and improvement of the proposed algorithm and system, considering emerging technologies or evolving requirements in the field of forest fire detection.

Answer:In future development and improvement of the proposed algorithms and systems, the following potential pathways can be considered to address the evolving needs of emerging technologies and the field of forest fire detection:

1. Introduction of deep learning technology: Deep learning technology has achieved great success in the field of computer vision, and it can be considered to be applied to forest fire detection. For example, the use of deeper convolutional neural networks (CNNS) or the use of emerging neural network architectures such as Transformer can be explored to improve the performance of algorithms.

2. Real-time monitoring and warning: With the development of technologies such as the Internet of Things and cloud computing, it is possible to consider combining fire detection systems with real-time monitoring and warning systems. In this way, fires can be detected in time and accurate warning information can be provided so that emergency measures can be taken.

3. Introduce automated decision support: In addition to fire detection, consider developing automated decision support systems to provide advice and decision support for fire response and fire fighting actions based on test results and other environmental information.

4. Consider edge computing and mobile platforms: With the development of edge computing and mobile computing platforms, it is possible to consider deploying fire detection algorithms and systems to edge devices or mobile platforms for faster response and wider coverage.

5. Continuous update of data sets and models: With the accumulation of data and the improvement of algorithms, the data sets used for training and evaluation can be continuously updated and expanded, and the model can be continuously improved and optimized to adapt to new fire scenarios and changing needs.

Round 2

Reviewer 2 Report

Comments and Suggestions for Authors

The authors responded to most of my comments and made appropriate corrections to the text of the manuscript. I believe that the manuscript can now be published.

Reviewer 4 Report

Comments and Suggestions for Authors

Most of comments have been considered in the revised version. The revised version is better than original submission in terms of paper organization and technical details. 

Reviewer 5 Report

Comments and Suggestions for Authors

Accept , All revision modified.